# Safety and immunogenicity of a SARS-CoV-2 Gamma variant RBD-based protein adjuvanted vaccine used as booster in healthy adults

Karina A. Pasquevich [1,2] ✉, Lorena M. Coria [1,2], Ana Ceballos[3], Bianca Mazzitelli[3], Juan Manuel Rodriguez[4], Agostina Demaría[1,2], Celeste Pueblas Castro[1,2], Laura Bruno[1,2], Lucas Saposnik[1,2], Melina Salvatori[3], Augusto Varese[3], Soledad González[5], Veronica V. González Martínez [5], Jorge Geffner[3], Diego Álvarez[1,2], Laboratorio Pablo Cassará R&D and CMC for ARVAC CG consortium*, Ethel Feleder[6], Karina Halabe[6], Pablo E. Perez Lera[6], Federico Montes de Oca[7], Julio C. Vega[7], Mónica Lombardo[8], Gustavo A. Yerino[6], Juan Fló [7] & Juliana Cassataro [1,2] ✉

A Gamma Variant RBD-based aluminum hydroxide adjuvanted vaccine called ARVAC CG was selected for a first in human clinical trial. Healthy male and female participants (18-55 years old) with a complete COVID-19-primary vaccine scheme were assigned to receive two intramuscular doses of either a low-dose or a high-dose of ARVAC CG. The primary endpoint was safety. The secondary objective was humoral immunogenicity. Cellular immune responses were studied as an exploratory objective. The trial was prospectively registered in PRIISA.BA (Registration Code 6564) and ANMAT and retrospectively registered in ClinicalTrials.gov (NCT05656508). Samples from participants of a surveillance strategy implemented by the Ministry of Health of the Province of Buenos Aires that were boosted with BNT162b2 were also analyzed to compare with the booster effect of ARVAC CG. ARVAC CG exhibits a satisfactory safety profile, a robust and broad booster response of neutralizing antibodies against the Ancestral strain of SARS-CoV-2 and the Gamma, Delta, Omicron BA.1 and Omicron BA.5 variants of concern and a booster effect on T cell immunity in individuals previously immunized with different COVID-19 vaccine platforms.

The severe acute respiratory virus 2 (SARS-CoV-2) was first identified in November 2019 and soon thereafter, emerging new viral variants dramatically impacted the dynamics of Coronavirus disease 2019 (COVID-19) spread, globally. These virus variants are, in general, more contagious than previous strains. Moreover, some variants are capable of immunological and/or therapeutic escape[1,2].

Numerous vaccines have been developed and proved effective to protect against severe disease, hospitalization, and fatal outcomes[3–5]. In Argentina, several platforms of COVID-19 vaccines have been introduced, including inactivated vaccines (BBIBP-CorV), viral vectored vaccines (Sputnik V, AZD1222, CanSino), and mRNA vaccines (BNT162b2 and mRNA-1273), resulting in a significant coverage of the

A full list of affiliations appears at the end of the paper. *A list of authors and their affiliations appears at the end of the paper.
✉e-mail: kpasquevich@iib.unsam.edu.ar; jucassataro@iib.unsam.edu.ar

population with complete primary vaccine series[6,7]. However, due to waning immunity and emergence of highly transmissible immune escape viral variants, two-dose COVID-19 vaccination programs may have not been enough to prevent breakthrough infections caused by these variants[8,9]. Clinical studies suggest that boosting with variant-adapted vaccines would optimize vaccine efficacy (VE) inducing strong and broad immune responses[10–12].

The pandemic has disproportionately affected low- and middle-income countries (LMICs), which make up about 85% of the world population. Therefore, pandemic remains a threat unless most people get vaccinated. In response to the constraints imposed by the COVID-19 pandemic, and the limited access of many Latin American countries to costly vaccines, Laboratorio Pablo Cassará S.R.L., an Argentinian pharmaceutical company, launched a vaccine development program against SARS-CoV-2. ARVAC CG is a receptor binding domain (RBD)-based protein aluminum hydroxide-adjuvanted vaccine candidate that was designed and produced entirely in Argentina to be used as booster or primary vaccine against SARS-CoV-2. The vaccine is a variant-adapted vaccine based on the highly immune evasive Gamma SARS-CoV-2 variant of concern (VOC). Non-clinical studies of this vaccine prototype in mice indicated that the Gamma-variant vaccine-candidate is more immunogenic and induces a broader nAb response than the Ancestral vaccine-candidate[13].

In this interim report safety and immunogenicity data after a booster dose of ARVAC CG vaccine from an ongoing first in human phase 1 study are presented. ARVAC CG exhibits a satisfactory safety profile, a robust and broad booster response of neutralizing antibodies against the Ancestral strain of SARS-CoV-2, the Gamma variant, and other VOCs (Delta, Omicron BA.1 and Omicron BA.5) and a booster effect on T cell immunity when used as booster in individuals previously immunized with different COVID-19 vaccine platforms.

## Results

### Population characteristics and local and systemic adverse events

The flowchart of the study is shown in Fig. 1. Demographic characteristics of the participants are presented in Table 1. Vaccination with ARVAC CG was well tolerated, with mild-to-moderate reactogenicity profiles (Fig. 2a). Overall, solicited local adverse events (AEs) were more frequently noticed after the first dose of the vaccine than after the second. At least one local AE was observed in 68.3% volunteers of Group A (25 μg vaccine) after the first administration and in 47.5% following the second ($p = 0.026$). In group B (50 μg vaccine), 60.0% of volunteers reported at least one local AE after the first injection whereas 27.8% following the second ($p = 0.0585$) (Supplementary Table 1).

The most frequent local AEs were discomfort/tenderness and pain at injection site in both groups (Fig. 2a). All reactions were transient and did not present complications.

Systemic AEs were less common. Reported frequency of solicited/unsolicited systemic AEs, was not significantly different between the first and second doses for both groups. In Group A, 33.3% and 35.6% of the volunteers reported at least one systemic AE after the first- and the second administration, respectively ($p = 0.848$), while in Group B, 40.0% and 22.2% of the volunteers reported at least one systemic AE after each dose, respectively ($p = 0.485$) (Supplementary Table 1).

The most frequent systemic AEs were drowsiness, headache, myalgia, and fatigue (Fig. 2b). Only one case of fever (38.3 °C) lasting one day was reported. Of note, 89.9% of reported AEs were grade 1 and there was no grade 3 or more severe AE (Supplementary Table 1).

No abnormal laboratory values were reported to be clinically significant. There were no serious AEs, deaths, or withdrawals due to an AE during the study. No cases of Guillain-Barré syndrome, thromboembolic events, myocarditis or pericarditis, or other AE of special interest have been reported.

### Immunogenicity results

After 14 days of the first ARVAC CG booster administration a significant increase in the nAb titers against all the five SARS-CoV-2 VOCs analyzed was observed in both ARVAC CG vaccine cohorts, compared to pre-vaccination titers ($P < 0.0001$) (Fig. 3). ARVAC CG 25 μg dose induced a 12.6 (95% CI, 8.8–17.9), 16.6 (95% CI, 11.8–23.4), 11.3 (95% CI, 7.8–16.5), 12.8 (95% CI, 9.2–18.0), and 8.6 (95% CI, 6.1–12.0) geometric mean fold rise (GMFR) in nAb titers against the Ancestral (Wuhan), Gamma, Delta, Omicron BA.1 or Omicron BA.5 variants of SARS-CoV-2, respectively (Fig. 3a). Additionally, immunization with the 50 μg dose induced a GMFR of 29.9 (95% CI, 12.6–70.6), 30.9 (95% CI, 13.4–71.5), 18.4 (95% CI, 8.2–41.1), 29.9 (95% CI, 13.0–68.3), and 13.0 (95% CI, 6.0–28.4) in nAb titers against Ancestral, Gamma, Delta, Omicron BA.1 or Omicron BA.5 variants of SARS-CoV-2, respectively (Fig. 3b). Of note, GMFR of nAb titers against Ancestral and Omicron BA.1 variants were significantly higher in Group B than in Group A ($P = 0.0448$ and $P = 0.0271$, respectively) (Supplementary Table 2).

Seroconversion rates were evaluated as the percentage of subjects with at least a fourfold increase (4×-seroconversion rates) or a tenfold increase (10×-seroconversion rates) in the nAb titers at a specific timepoint respect to baseline values. After 14 days of a booster with ARVAC CG the 4×-seroconversion rates for the Ancestral SARS-CoV-2 strain were 88.3 % (95% CI, 77.8–94.2) in the 25 μg cohort and 90.0% (95% CI, 69.9–98.2) in the 50 μg cohort. Whereas the corresponding 4×-seroconversion rates were 90.0% (95% CI, 79.8–95.3) and 85% (95% CI, 64.0–94.8) against the Gamma VOC, 80.0% (95% CI, 68.2–88.2) and 85.0% (95% CI, 64.0–94.8) for Delta VOC, 93.3% (95% CI, 84.1–97.4) and 85.0% (95% CI, 64.0–94.8) for Omicron BA.1 VOC, and 80.0% (95% CI, 68.2–88.2) and 80.0% (95% CI, 58.4–91.9) for Omicron BA.5 VOC, respectively (Supplementary Table 2).

Fourteen days after boosting, 4×-seroconversion rates for all tested variants were similar in both dosage groups, while 10×-seroconversion rates for Omicron BA.1 VOC were significantly higher in the 50 μg cohort (Supplementary Table 2).

Based on pre-existing anti-N IgG titers and/or previous history of COVID-19, individuals were stratified into two populations: seronegative individuals with no previous COVID-19 history and seropositive and/or with previous history of COVID-19 individuals. Both populations developed similar nAb GMTs against Ancestral, Gamma, Delta, Omicron BA.1 and Omicron BA.5 after 14 days of ARVAC CG booster. Moreover, GMFR from baseline were similar for both populations in groups A and B either when analyzed in all individuals of each group (Supplementary Fig. 1) or in the subgroup of subjects who received BBIBP-CorV as primary vaccination (Supplementary Fig. 2).

No significant differences were found in the nAb responses and seroconversion rates between female or male volunteers, except for the 4×-seroconversion rate in nAbs against Delta VOC (Supplementary Table 3). In addition, no differences in the nAb titers after booster were observed in the ARVAC CG cohorts when participants were stratified regarding their time since completion of primary vaccination series (time < 180 days vs. ≥ 180 days) (Supplementary Fig. 3).

When volunteers were subdivided regarding the primary vaccination received, a significant increase in the nAb GMTs against the different viral variants was observed in all previously vaccinated subgroups (Fig. 4). Comparisons between group A and B in terms of GMFR of nAb titers and seroconversion rates was also assessed in the subgroup of individuals that had received the BBIBP-CorV vaccine as primary vaccination scheme. The GMFR in nAb titers against Ancestral, Gamma, Omicron BA.1 and Omicron BA.5 VOCs were significantly higher in Group B than in Group A ($P = 0.0154$, $P = 0.0118$, $P = 0.0160$, and $P = 0.0459$, respectively). Fourteen days after boosting, 4×-seroconversion rates for all tested variants were similar in both groups, whereas the 10×-seroconversion rates for the Ancestral and Omicron BA.1 VOC were significantly higher in the 50 μg cohort than in the 25 μg cohort (Supplementary Table 3).

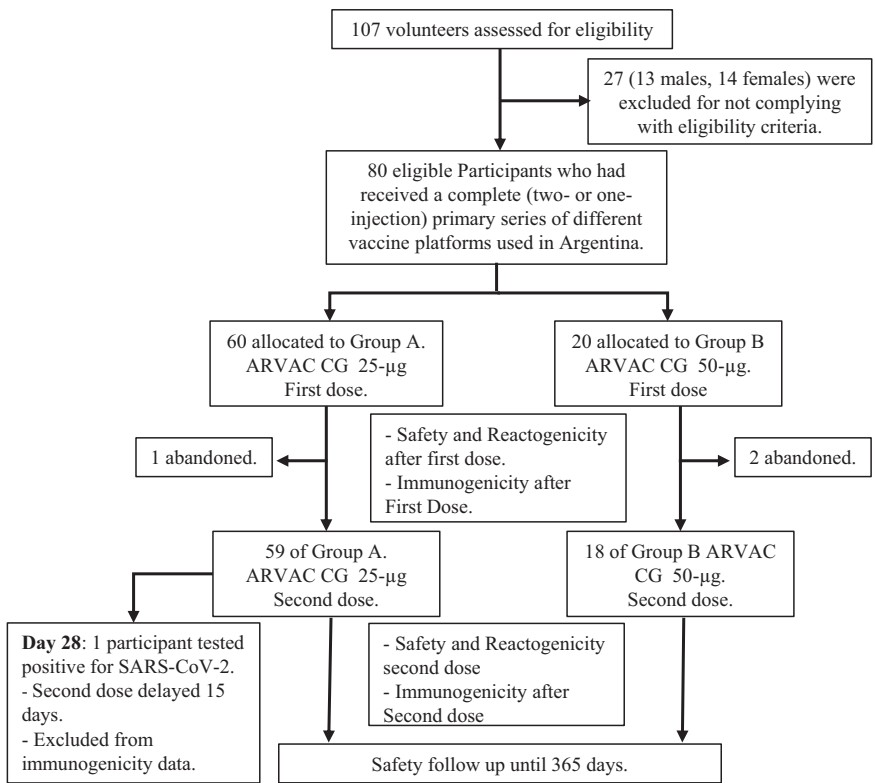

**Fig. 1 | Subject disposition in Phase-1 study (consort diagram).** Trial profile showing the study groups, number of participants, and vaccine dose received.

**Table 1 | Demographic Characteristics of the Participants in the ARVAC-F1-001 Trial at Enrolment**

| Variable | | ARVAC CG 25 µg Group A | ARVAC CG 50 µg Group B | Total |
|---|---|---|---|---|
| No. of participants | | 60 | 20 | 80 |
| Sex | Male (N (%)) | 29 (48.3) | 8 (40) | 37 (46.3) |
| | Female (N (%)) | 31 (51.7) | 12 (60) | 43 (53.8) |
| Age (Median (IQRª)) | | 32 (25–37.25) | 27 (20.75–33.25) | 31 (24–37) |
| Body-mass index (Median (IQR)) | | 25.7 (22.9–27.8) | 26.1 (21.1–27.4) | 25.8 (22.9–27.8) |
| COVID-19 Primary Vaccine Platform | BBIBP-CorV (N (%)) | 20 (33.3) | 14 (70.0) | 34 (42.5) |
| | Sputnik V (rAd26/rAd5) (N (%)) | 21 (35.0) | 1 (5.0) | 22 (27.5) |
| | ChAdOx1-S (N (%)) | 17 (28.3) | 1 (5.0) | 18 (22.5) |
| | CanSino (Ad5-nCoV) (N (%)) | 1 (1.7) | 0 (0) | 1 (1.3) |
| | Janssen (Ad26.CoV2.S) (N (%)) | 0 (0) | 1 (5.0) | 1 (1.3) |
| | Heterologous schedule (N (%)) | 1 (1.7) | 3 (15.0) | 4 (5.0) |
| Time (months) since last dose of primary immunization schedule. (Median (IQR)) | | 7.9 (6.2–8.7) | 6.6 (4.9–8.1) | 7.7 (6.0–8.4) |
| Prior COVID-19[b] | No (N (%)) | 55 (91.7) | 16 (80.0) | 71 (88.8) |
| | Yes (N (%)) | 5 (8.3) | 4 (20.0) | 9 (11.3) |
| | Time (months) since infection (Median (IQR)) | 12.0 (4.9–20.3) | 5.7 (5.6–5.9) | 5.8 (5.3–12.0) |
| Seropositive for N (SARS-CoV-2 nucleoprotein)-specific IgG | No (N (%)) | 23 (38.3) | 9 (45.0) | 32 (40.0) |
| | Yes (N (%)) | 37 (61.7) | 11 (55.0) | 48 (60.0) |

[a]*IQR* Interquartile range.
[b]Confirmed diagnostic of COVID-19 (PCR, Antigen test or by epidemiological diagnostic).

Similar results were observed at 28 days post-booster vaccination with ARVAC CG (Supplementary Fig. 4 and Supplementary Table 4). Levels of anti-RBD and anti-spike antibodies raised significantly after 28 days with respect to baseline levels in both study groups (Supplementary Fig. 5).

To preliminarily assess the comparative immunogenicity of ARVAC CG, nAb GMTs and GMFR were compared with those in samples from a cohort of individuals out of the protocol, with similar demographic characteristic (Table 2), who had received a heterologous booster dose with the Ancestral-based BNT162b2 mRNA vaccine. Baseline nAb GMTs against the five viral variants in both ARVAC CG cohorts were similar to those observed before the booster with BNT162b2 ($P > 0.05$). After 14 or 28 days the nAb GMTs against the Ancestral SARS-CoV-2 in group A were similar to that achieved after

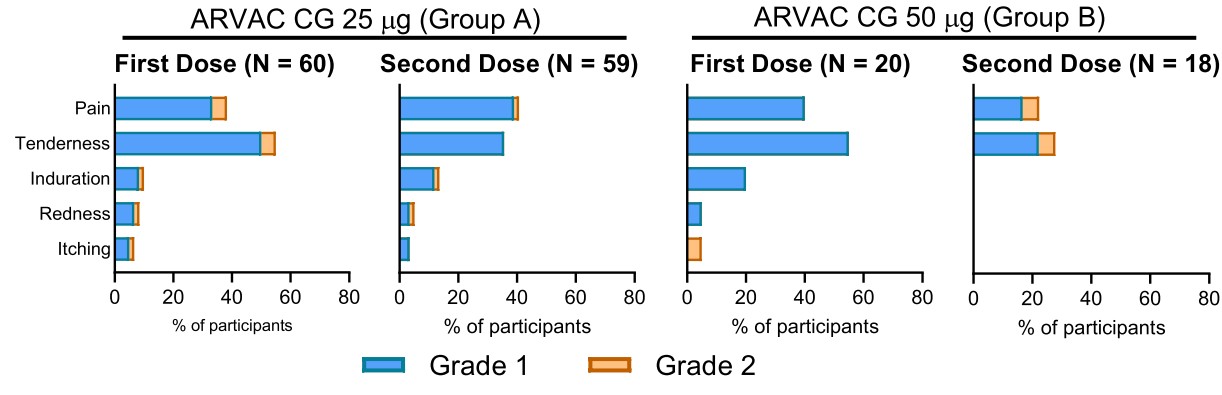

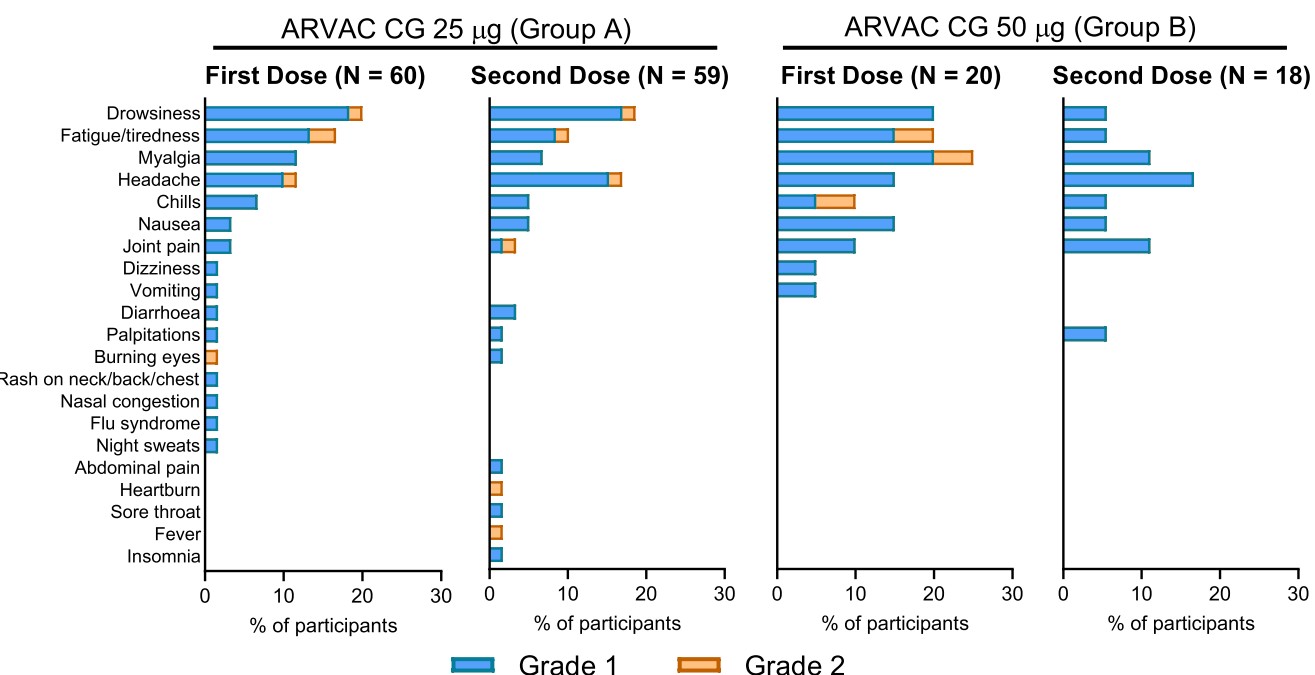

**Fig. 2 | Safety profile. a** Percentage of participants in each study group with the indicated injection site AEs up to 7 days after the first or the second injection. **b** Percentage of participants in each study group with the indicated systemic AEs recorded up to 28 days after each vaccine administration. Events were classified according to the FDA toxicity grading scale for healthy adult and adolescent volunteers enrolled in preventive vaccine clinical trials (Mild (Grade 1), Moderate (Grade 2), Severe (Grade 3), Potentially Life Threatening (Grade 4))[28].

BNT162b2 booster. However, nAb GMTs against Gamma, Delta, Omicron BA.1, and Omicron BA.5 were significantly higher in group A than those reached in BNT162b2 boosted individuals. Similarly, in group B nAb GMTs against Gamma VOC after 14 days of booster and against Omicron BA.1 and Omicron BA.5 VOCs at all tested timepoints were significantly higher than the corresponding nAb GMTs in BNT162b2 boosted individuals (Fig. 5a−e).

While GMFR in nAb titers against Ancestral, Gamma and Delta VOCs after a booster dose with BNT162b2 were similar to those elicited in the ARVAC CG cohorts ($P > 0.05$), GMFR of nAb titers against Omicron BA.1 and BA.5 were significantly lower in the BNT162b2- than in the ARVAC CG boosted individuals (Fig. 5f). Similar results were obtained when the nAb responses of the BNT162b2 group were compared to those of ARVAC CG boosted individuals whose time since completion of primary vaccination series and booster was less than 180 days (Supplementary Fig. 6) or when compared only the

individuals whose primary vaccination scheme was rAd26/rAd5 (Sputnik V vaccine) (Supplementary Fig. 7).

Transformation of nAb titers to IU/ml, allowed the evaluation of the percentage of participants with nAb levels associated with high VE. In group A, rates of participants with nAb levels higher than of 949 IU/ml before the booster were 26.7% (95% CI 39.0, 17.1), 3.3% (95% CI 11.4, 0.6), 6.7% (95% CI 15.9, 2.6), and 1.7% (95% CI 8.9, 0.1) for the Ancestral, Gamma, Delta, and Omicron BA.1 SARS-CoV-2 VOCs, respectively. These rates rose significantly after 14 days of having received the booster to reach 83.3% (95% CI 90.7, 72.0), 80.0 % (95% CI 88.2, 68.2), 60.0% (95% CI 71.4, 47.4), and 78.3% (95% CI 86.9, 66.4) for each viral variant, respectively. Likewise, in group B the proportion of participants with nAb levels higher than 949 IU/ml raised significantly from 15.0% (95% CI 5.2, 36.0), 0.0% (95% CI 0.0, 16.1), 5.0% (95% CI 0.3, 23.6), and 10.0% (95% CI 1.8, 30.1) for the Ancestral, Gamma, Delta, and Omicron BA.1 SARS-CoV-2 VOCs respectively at baseline, to 80.0%

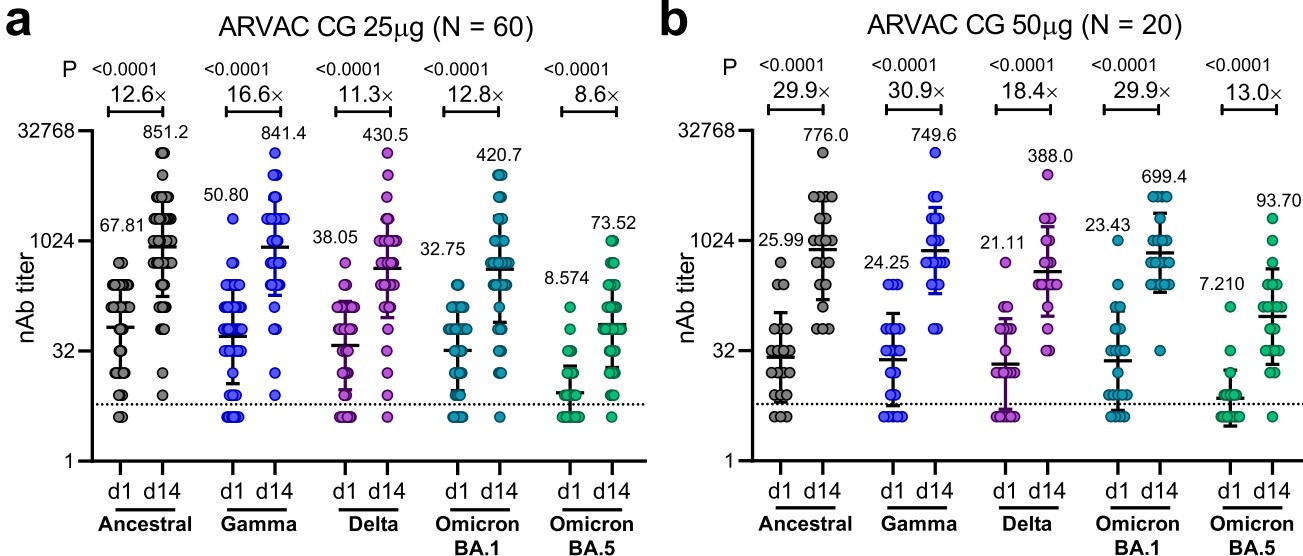

**Fig. 3 | Administration of ARVAC CG booster increases the nAb titers against the Ancestral, Gamma, Delta, Omicron BA.1 and Omicron BA.5 variants of SARS-CoV-2.** The nAb titers against the Ancestral, Gamma, Delta and Omicron BA.1 and Omicron BA.5 variants of SARS-CoV-2 in plasma samples of individuals boosted with ARVAC CG 25 μg (*N* = 60 individuals) (**a**) or 50 μg (*N* = 20 individuals) (**b**) prior to the vaccine administration (d1) or after 14 days of booster administration (d14). Each point represents the nAb titer of a volunteer at the indicated time point and against the depicted viral variant. The nAb geometric mean titers (GMTs) and 95% CIs are shown as horizontal and error bars, respectively. The numbers depicted

above the individual points for each specified time point and viral variant represent de GMT. The fold increase in the GMT from day 1 to day 14 (GMFR) for each specified variant are shown with a number followed by a ×. The dashed line represents the positivity threshold on the virus neutralization assay. Statistical differences were analyzed using the two-tailed Wilcoxon pair-matched test. *P*-values are depicted above the data sets that were compared. In panel (**a**) all *P* values were smaller than 10e-15; in panel (**b**) *P* = 0.000004 for Ancestral strain, Gamma and Omicron BA.1 VOCs, *P* = 0.000099 for Delta VOC and *P* = 0.000015 for Omicron BA.5.

(95% CI 58.4, 91.9), 70.0% (95% CI 48.1, 85.5), 45.0% (95% CI 25.8, 65.8), and 95.0% (95% CI, 76.4, 99.7) for each viral variant, respectively (Supplementary Fig. 8).

A significant increase in the frequency of IFN-γ producing cells upon in vitro re-stimulation with SARS-CoV-2 RBD peptide pools was observed in both ARVAC CG cohorts in comparison with the levels observed before the booster. In addition, a slight increase of IL-4 producing cells was observed after the booster in group A participants (Fig. 6). Interestingly, booster vaccination with 25 μg or 50 μg of ARVAC CG led to increases in antigen-specific cellular immune responses in individuals primed with different vaccine platforms (Supplementary Fig. 9).

A second booster with ARVAC CG was given to the volunteers after 28 days to collect safety data after two vaccine administrations. The nAb titers remained significantly higher than baseline values (d1) after 42 days and 56 days of first dose administration (Supplementary Fig. 10).

## Discussion

The main findings of this study are that the vaccine candidate ARVAC CG when given as a booster dose is well tolerated and induces a robust and broad nAb response against several SARS-CoV-2 VOCs.

The interim results of this phase 1 study indicate that ARVAC CG vaccine given to individuals who previously received a complete primary vaccination regimen has a clinically acceptable safety and reactogenicity profile for both antigen doses (25 μg and 50 μg).

Immunogenicity results indicate that ARVAC CG as a booster dose induces a sharp increase of broadly nAb titres against the Ancestral SARS-CoV-2 strain, the Gamma SARS-CoV-2 VOC, which is the vaccine prototype strain, as well as against antigenic distant SARS-CoV-2 VOCs like Delta, Omicron BA.1 and Omicron BA.5. Moreover, the ARVAC CG 50 μg dose outperformed the 25 μg dose in terms of nAb titers reached against the Ancestral and Omicron BA.1 viral variants and 10×-seroconversion rates in nAb against Omicron BA.1. The booster effect

after one dose of ARVAC CG vaccine was evident despite the variety of immunization schemes received by the study participants.

The differences in the proportions of subjects with different types of primary vaccine regimen in the low-dose and the high-dose groups may difficult comparisons. However, the possibility to include different primary vaccination schemes was important in this phase 1 study to have a representation of the diversity in primary vaccination schemes that were used in Argentina. Since most individuals in group B had received the BBIBP-CorV vaccine as primary vaccination regimen and there were approximately equal numbers of BBIBP-CorV recipients in the high and low dose groups, the comparison of these subgroups was also performed. Similar to the findings when all volunteers were included, in these more homogeneous subpopulations, the 50 μg dose was consistently more immunogenic than the 25 μg dose. Although the time since prior COVID-19 declared by participants was quite different between the high and low dose groups and this could be a limitation, the analysis of anti-N in sera of all individuals indicated that both populations were similar in their inferred previous exposure to the virus and that the immune responses after vaccination are independent of previous infection status.

With the emergence of new VOCs it is clear that breakthrough infections can occur in vaccinated persons, including those with previous SARS-CoV-2 infection[14,15]. Therefore, the capability to boost the immune responses in individuals with a previous history of infection becomes critical, enhancing protection against COVID-19 and post-COVID conditions[16]. The results of this study highlight the potential benefit of the ARVAC CG vaccine for all populations regardless of their prior COVID-19 serological status.

The performance of the ARVAC CG booster dose regarding nAb GMTs and GMFR, was similar to that of the BNT162b2 booster against Ancestral and Delta VOCs, but better for Gamma, Omicron BA.1 and BA.5 VOCs. Similarly, booster shots with Beta variant-based vaccines elicit broad nAb responses against the Ancestral, the Beta and the Omicron BA.1 VOC, which were higher than that elicited by a booster

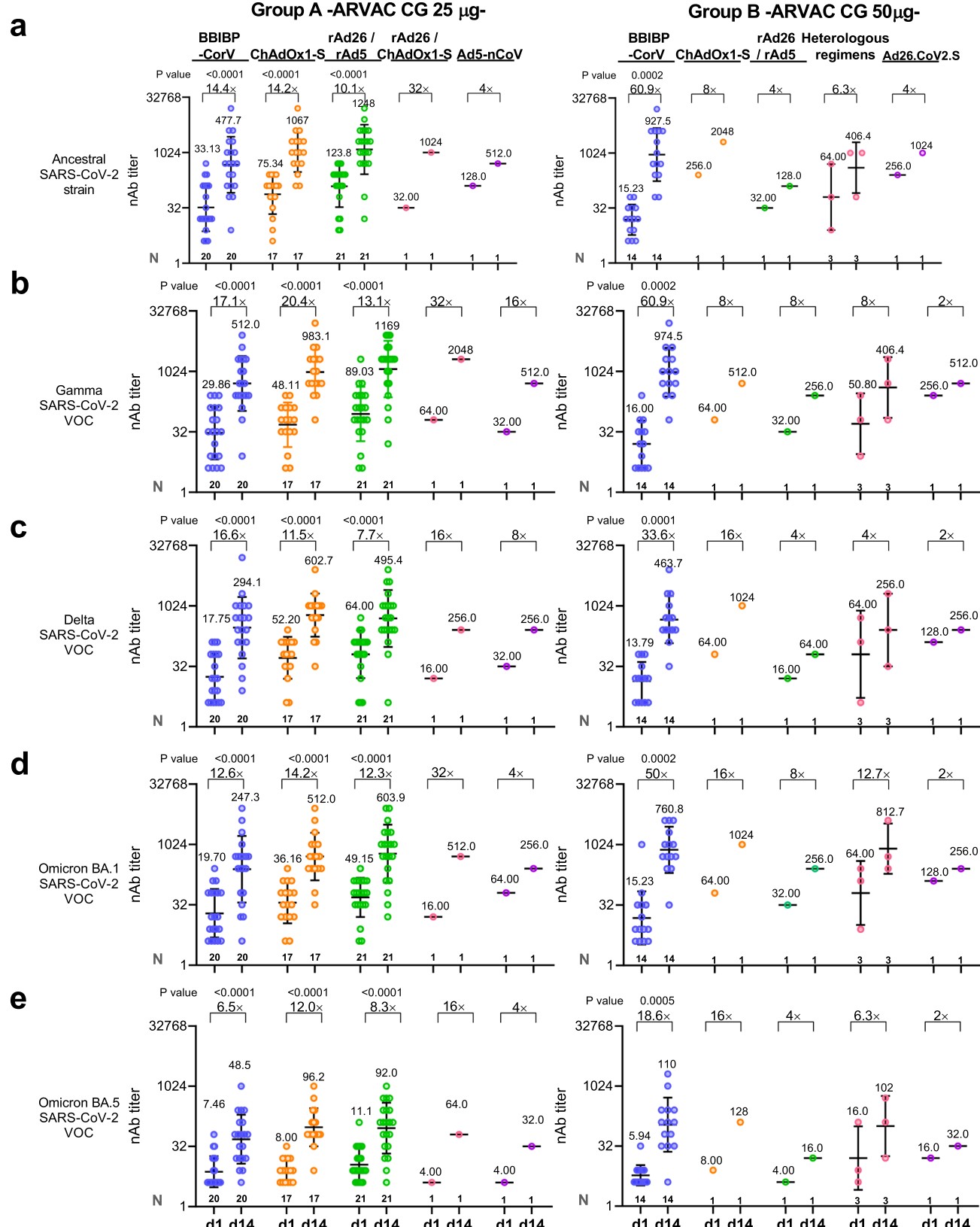

with an ancestral-variant based vaccine[10,17,18]. Results presented here suggest that a booster with a Gamma variant RBD-based vaccine increases the breadth of the nAb and are in agreement with non-clinical results of this vaccine formulation[13].

Even though there are not well-established threshold values of nAb levels that correlate with protection against symptomatic SARS-

CoV-2 infection, an increasing number of studies using standardized methods offer hints about the nAb levels that might be associated with protection[15,19–21]. In this regard, nAb levels of ~100-120 IU/ml have been correlated with ~80% VE against symptomatic infection, whereas nAb levels of ~900-1030 IU/ml correlate with ~90% VE against symptomatic infection[19]. Results presented here suggest that a booster with ARVAC

**Fig. 4 | Administration of ARVAC CG as booster increases the nAb titers against the Ancestral, Gamma, Delta, Omicron BA.1 and Omicron BA.5 variants of SARS-CoV-2 in individuals with different primary vaccinations schemes.** Neutralizing antibody titers against the Ancestral (**a**), Gamma (**b**), Delta (**c**), Omicron BA.1 (**d**) and Omicron BA.5 (**e**) variants of SARS-CoV-2 in plasma samples of individuals boosted with ARVAC CG 25 μg (left panels) or 50 μg (right panels) prior to the vaccine administration (d1) or after 14 days of booster (d14). Participants in each cohort were grouped according to the primary vaccination scheme received. ARVAC CG 25 μg cohort: BBIBP-CorV ($N = 20$), ChAdOx1-S ($N = 17$), rAd26/rAd5 ($N = 21$), rAd26/ChAdOx1-S ($N = 1$) and Ad5-nCoV ($N = 1$). ARVAC CG 50 μg cohort: BBIBP-CorV ($N = 14$), ChAdOx1-S ($N = 1$), rAd26/rAd5 ($N = 1$), heterologous vaccination regimens (ChAdOx1-S/mRNA1273 or BBIBP-CorV/BNT162b2) ($N = 3$) and Ad26.CoV2.S ($N = 1$). Each point represents the nAb titer of a volunteer.

In subgroups with $N > 1$, the nAb GMTs with geometric SD are shown as horizontal and error bars, respectively. The numbers depicted above the individual points for each specified time point represent the GMT values. The GMFR from day 1 to day 14 for each specified variant are shown with a number followed by a ×. The number of participants included in each data set analyzed are depicted in the bottom of each data set (N). Statistical differences were performed using the two-tailed Wilcoxon pair-matched test. $P$-values are depicted above the data sets that were compared. Exact $P$-values (d1 vs. d14 nAb titers) in ARVAC CG 25 μg subgroups (left panels) are: BBIBP-CorV ($P = 0.000002$ (**a**, **b**, **d**), $P = 0.00003$ (**c**), and $P = 0.000004$ (**d**, **e**), ChAdOx1-S ($P = 0.000002$ (**a**, **b**, **c**) and $P = 0.00002$ (**d**, **e**). In ARVAC CG 50 μg BBIBP-CorV primary vaccination exact $P$-values are: $P = 0.0002$ (**a**, **b**, **d**), $P = 0.0001$ (**c**) and $P = 0.0005$ (**e**).

CG significantly increases the proportion of individuals with nAb titers that correlate with high VE.

T-cell immunity is crucial to combat acute SARS-CoV-2 infection and for the development of long-term immunity[22]. While antibody titers tend to wane rapidly and show limited neutralizing activity to newly arising VOCs, T-cell memory is largely conserved[22]. ARVAC CG boosted T cell immunity, which might contribute to eliminate virus-infected cells. Although it has been reported that durable spike-specific T-cell responses after different COVID-19 primary vaccination regimens are not further enhanced by an mRNA booster[23], results presented here show that ARVAC CG booster significantly increases the proportion of antigen-specific IFN-γ and IL-4 producing T cells in individuals previously vaccinated with different primary schemes.

One limitation of this study is the lack of randomization for volunteers. Nevertheless, despite the sequential study design, enrolment dates of the booster groups occurred within weeks of each other, representing similar epidemiologic environments of circulating variants. The comparison of the immunogenicity results of ARVAC CG cohorts with those of a contemporaneous BNT162b2 booster study has another limitation, since the study participants presented some demographic differences in their ages (36.5 in BNT162b2 group versus 32 and 27 years in groups A and B, respectively). Although the time from last SARS-CoV-2 vaccine dose (4.2 months in the BNT162b2 boosted subjects versus 7.9 and 6.6 months in group A and B, respectively) was different this did not influence the immune outcomes in this study. Peak of nAb titers reached after booster in both ARVAC CG cohorts were similar whether the time since primary vaccination completion to booster was short (<180 days) or long (≥180 days). Indeed, ARVAC CG boosters at short time (<180 days) showed a better performance than BNT162b2 booster. The confirmed previous SARS-CoV-2 infection history (0% in group BNT162b2 versus 8% and 20% in groups A and B, respectively) was quite different, nevertheless anti-N antibodies serology indicated that the three populations had similar proportions of seropositive individuals and might had similar previous contact with the virus. The proportion of primary vaccination schemes was also different, since most individuals in BNT162b2 boosted cohort had received the rAd26/rAd5 (Sputnik V) vaccine as primary vaccination regimen, however the comparison among subjects with the same primary vaccination led to similar results.

In this phase 1 study, safety was demonstrated after two ARVAC CG administrations highlighting that ARVAC CG is safe. Immune responses after a single booster dose could be assessed only after 14 and 28 days of administration. After the second ARVAC CG administration the nAb remained significantly higher than baseline but no booster effect was observed. The lack of booster effect may be due to the short interval between boosters, that may not be the optimal in terms of immunological performance[24-26]. Longer-term follow-up of immune responses after a single booster dose will have to be studied in an ongoing phase 2/3 study.

While both formulations of ARVAC CG exhibited a favorable safety and reactogenicity profile eliciting broadly nAb responses and T cell immunity against SARS-CoV-2, the 50 μg dose outperformed the 25 μg dose in certain of the immunogenicity variables evaluated. Therefore, the 50 μg dose vaccine is currently being tested in an ongoing Phase 2/3 study to evaluate its safety, and immunogenicity in a larger population. Selection of the 50 μg vaccine dose allowed the testing of a bivalent vaccine containing 25 μg of Gamma-based antigen plus 25 μg of Omicron BA.5-based antigen to improve the response capacity, as part of a Phase 2/3 study, in accordance with the recommendations from the advisory committee on immunization practices for the use of bivalent booster doses of COVID-19 vaccines to increase protection against circulating VOCs and to broaden neutralization to previous and potentially yet-to-emerge variants[27].

There is a need for widespread immunization programs with new generation of COVID-19 booster vaccines which provide a wide breadth of protection against constantly emerging SARS-Cov-2 variants. Nonetheless, LMICs lag far behind in this effort due to limitations in affordability and accessibility to vaccines. Hence, developments such as ARVAC CG may offer an opportunity to overcome some of these challenges and improve the response capacity of many countries, worldwide.

## Methods

### Trial design and oversight

The trial was prospectively registered in PRIISA.BA (Registration Code 6564) and in ANMAT. Registration was performed in February 2022, before the study beginning and before enrollment of the first

**Table 2 | Demographic Characteristics of the Participants of the surveillance strategy implemented by the Ministry of Health of the Province of Buenos Aires that were boosted with BNT162b2 at Enrolment**

| Variable | | |
|---|---|---|
| No. of participants | | 18 |
| Sex | Male (N (%)) | 5 (27.8) |
| | Female (N (%)) | 13 (72.2) |
| Age (Median (IQR[a])) | | 36.5 (32–45) |
| COVID-19 Primary Vaccine Platform | BBIBP-CorV (N (%)) | 2 (11.1) |
| | Sputnik V (rAd26/rAd5) (N (%)) | 16 (88.9) |
| | Time since last dose of primary immunization schedule (in months, (Median (IQR)) | 4.2 (3.8–4.6) |
| Prior COVID-19[b] | No (N (%)) | 18 (100.0) |
| | Yes (N (%)) | 0 (0.0) |
| Seropositive for N (SARS-CoV-2 nucleoprotein)-specific IgG | No (N (%)) | 9 (50.0) |
| | Yes (N (%)) | 9 (50.0) |

[a]IQR Interquartile range.
[b]Confirmed diagnostic of COVID-19 (PCR, Antigen test or by epidemiological diagnostic).

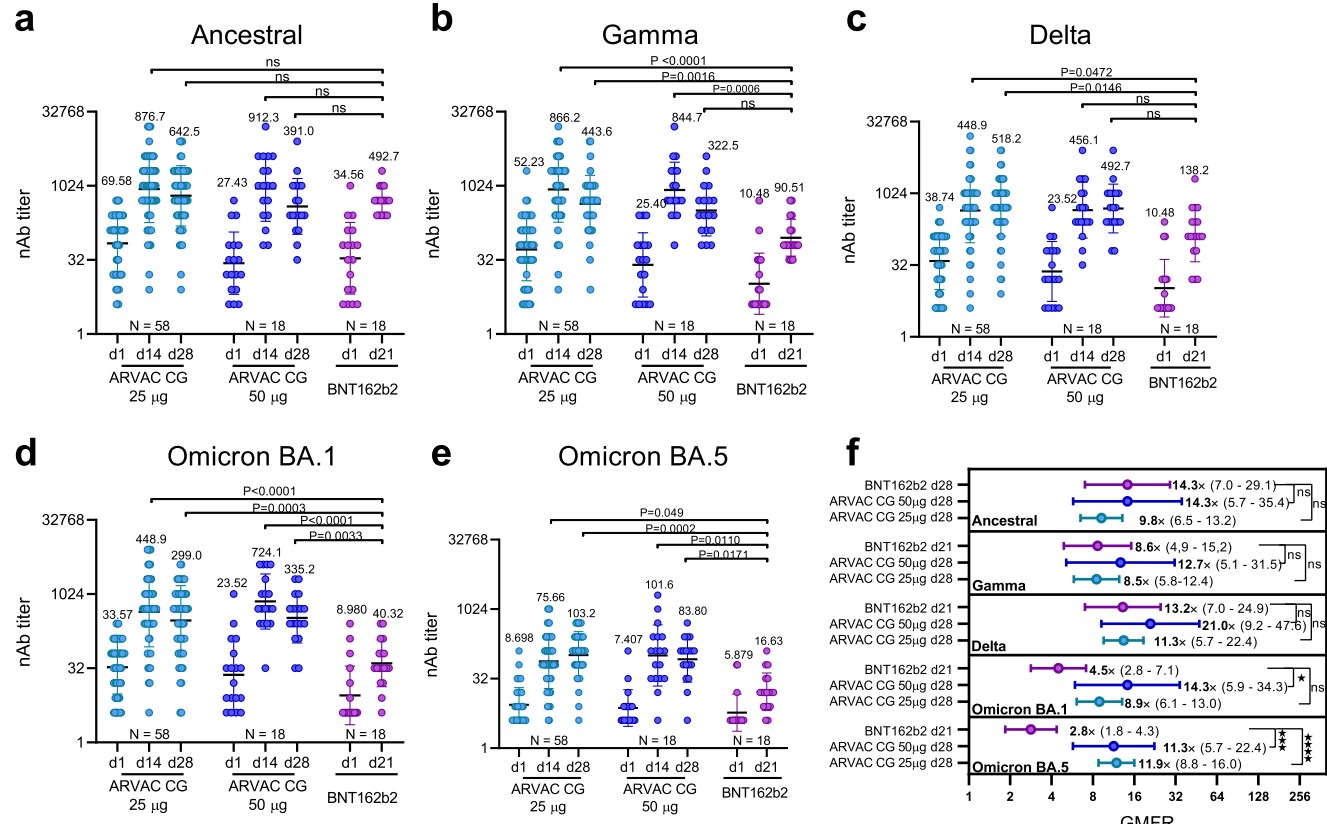

**Fig. 5 | Comparison of nAb GMT and GMFR after booster with ARVAC CG or booster with BNT162b2.** Neutralizing antibody titers against the Ancestral (**a**), Gamma (**b**), Delta (**c**), Omicron BA.1 (**d**) and Omicron BA.5 (**e**) variants of SARS-CoV-2 in plasma samples of individuals boosted with the indicated vaccine (ARVAC CG 25 μg, ARVAC CG 50 μg or BNT162b2) prior booster administration (d1) or at the indicated days after booster administration (d14, d21, d28). Each point represents the nAb titer of a volunteer. Data are from participants with no missing data at all analyzed time points (ARVAC CG 25 μg (*N* = 58), ARVAC CG 50 μg (*N* = 18) or BNT162b2 (*N* = 18). The nAb GMTs and 95% CIs are shown as horizontal and error bars, respectively. The numbers depicted above the individual points for each specified time point and viral variant represent the GMTs. The number of participants included in each data set analyzed is depicted in the bottom of the graph (*N* = number of individuals in each data set). Statistical differences were analyzed using the Kruskal-Wallis test followed by the Dunn´s multiple comparison test.

*P*-values are depicted above the data sets that were compared. ns: *P* > 0.05. Exact *P*-values for each comparison are: BNT162b2 d21 vs. ARVAC CG 25 μg d14: *P* = 0.000007 (**b**), *P* = 0.0472 (**c**), *P* = 0.00001 (**d**) and *P* = 0.049 (**e**); BNT162b2 d21 vs. ARVAC CG 25 μg d28: *P* = 0.0016 (**b**), *P* = 0.015 (**c**), *P* = 0.0003 (**d**) and *P* = 0.0002 (**e**); BNT162b2 d21 vs. ARVAC CG 50 μg d14: *P* = 0.0006 (**b**), *P* = 0.00003 (**d**) and *P* = 0.0110 (**e**); BNT162b2 d21 vs. ARVAC CG 50 μg d28: *P* = 0.0033 (**d**) and *P* = 0.0171 (**e**). **f** Fold increases in the GMT from day 1 to day 21 or 28 (GMFR) for each specified variant represented by a point and written with a number followed by a ×. The horizontal lines represent the 95% CIs. Data are from participants with no missing data at baseline and at all time points analyzed (ARVAC CG 25 μg (*N* = 58), ARVAC CG 50 μg (*N* = 18) or BNT162b2 (*N* = 18). Statistical differences were analyzed using Kruskal-Wallis test followed by the Dunn´s multiple comparison test. ns: *P* > 0.05, *\*P* = 0.0243; \*\*\**P* = 0.0009; \*\*\*\**P* = 0.00004.

participant. Also the same protocol with no changes was retrospectively registered on December 23, 2022 in ClinicalTrials.gov (NCT05656508). The trial was conducted at Clinical Pharma (Clínica CIAREC, Intense Life S.A, Buenos Aires, Argentina). In this open-label, first-in-human, dose-escalation, phase 1 clinical trial, eligible volunteers were healthy men and nonpregnant women, aged 18 to 55, with a body-mass index of 18 to 30 and with a complete COVID-19 vaccine primary schedule. Health status, assessed during the screening period, was based on medical history and extensive clinical laboratory tests, vital signs, and physical examination. Participants with a history of SARS-Cov-2 infection or COVID-19 within 60 days prior to recruitment into the study, or who tested positive in real-time polymerase-chain-reaction (RT-PCR) assay at screening or worked in an occupation with high risk of exposure to SARS-CoV-2, as well as those with an incomplete COVID-19 vaccine primary schedule or who had received the last COVID19 primary vaccine shot within 4 months prior to recruitment into the study or have received a booster dose of any COVID-19 vaccine, were excluded. Participants sex (male, female) was assigned based on sex assigned at birth, as self-reported by the participant.

The study protocol was initiated on April 10, 2022. Twenty seven out of 107 volunteers (13 males, 14 females) were excluded for not complying with eligibility criteria. Participants were recruited between April 28 and June 23, 2022, and sequentially assigned to one of two vaccine groups, one receiving a 25 μg dose of ARVAC CG (Group A) and the other a 50 μg dose (Group B). A sequential assignment plan was prespecified in the study protocol. In the first stage of enrollment, the first five enrolled participants received the low dose vaccine (25 μg/dose). Only one participant per day was vaccinated. Afterwards in the second stage of enrollment, participants 6th to 10th received the high dose vaccine formulation (50 μg/dose). Only one participant per day was vaccinated. The next fifty-five enrolled participants received the 25 μg/dose (stage 3), and then the last fifteen participants received the 50 μg/dose (stage 4). Of the 80 volunteers who received at least one intramuscular dose of ARVAC CG in the deltoid, three were excluded 28 days after the first dose of the vaccine due to impossibility to complete the protocol for personal reasons. Of the 77 remaining, 59 were inoculated with two 25 μg vaccine doses, and 18 with two 50 μg vaccine doses. One study participant was tested as SARS-CoV-2 positive in the fifth visit of the protocol (day 28 after first dose). The

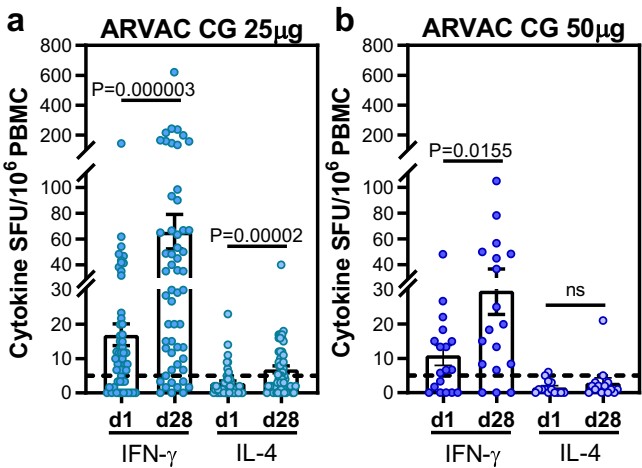

**Fig. 6 | ARVAC CG booster induces significant increase of Th1-predominant cell response measured by IFN-γ and IL-4 ELISpot after restimulation of PBMCs with RBD spanning peptide pool.** Before booster administration (d1) and after 28 days (d28) of administration of ARVAC CG 25 μg (**a**) or 50 μg (**b**) dose, RBD specific cellular responses were measured by IFN-γ and IL-4 ELISpot in PBMCs. Samples from 58 subjects and 18 subjects from ARVAC CG 25 μg and 50 μg cohorts were analyzed. Shown are spot-forming units (SFU) per $1 \times 10^6$ PBMCs producing IFN-γ and IL-4 after stimulation with RBD peptide pool from samples with viable cells: $N = 55$ for IFN-γ and $N = 51$ for IL-4 in ARVAC CG 25 μg group and $N = 18$ for IFN-γ and $N = 14$ for IL-4 in ARVAC CG 50 μg group. Each point represents the result from a subject. Bars indicate the mean, and error bars the SEM. Statistical analysis (d1 vs. d28) was performed by the two tailed Wilcoxon matched pairs signed rank test. **a** IFN-γ SFU d1 vs. d28: $P = 0.000003$; IL-4 SFU d1 vs. d28: $P = 0.00002$; (**b**) IFN-γ SFU d1 vs. d28: $P = 0.0155$; IL-4 SFU d1 vs. d28: $P > 0.05$.

participant was completely asymptomatic and following protocol instructions the application of the second dose was delayed. While the safety data of this volunteer at 28 days and after the second booster are included, the immunogenicity data at day 28 and at later time points were excluded because the immunogenicity against the virus may shape the antibody response and lead to misinterpretation of the results. Investigators, and laboratory personnel involved in assays were blind to assignment until the end of the follow-up period.

Missing data or deviations from original protocol were infrequent and inconsequential.

Safety was assessed according to the scheme established in the Guidance for Industry, Toxicity Grading Scale for Healthy Adult and Adolescent Volunteers Enrolled in Preventive Vaccine Clinical Trials[28]. Solicited local and systemic adverse events (AE) were recorded during the first 7 days after each dose of vaccine received, unsolicited events were recorded during the first 28 days after vaccination, laboratory tests were carried out after 7 and 28 days of each dose. The following symptom grading was used for local and systemic AE: grade 1 (mild) to grade 4 (potentially life-threatening). All safety information collected was available to an external Independent Committee of Data Review for continuous monitoring of any relevant AE and recommendations of modifying or interrupting the protocol as necessary.

Causality assessment of AE was based on the standard definition and application of terms for vaccine pharmacovigilance as stated by the Report of the CIOMS/WHO Working Group on Vaccine Pharmacovigilance[29].

The immunogenicity of ARVAC CG was compared to 18 samples obtained during the COVID-19 serology surveillance strategy implemented by the Ministry of Health of the Province of Buenos Aires, between February 4 and March 31, 2022. These were individuals with similar demographic characteristics who received a heterologous booster with the Ancestral-based BNT162b2 mRNA vaccine administered at least 3 months after a two-dose primary schedule (Table 2).

## Ethical statement
All participants provided written informed consent before enrolment in the trial and after the nature and possible consequences of the study were explained. The study was conducted according to the Declaration of Helsinki. The trial protocol was approved on March 09, 2022, by the Ethic Committee in Clinical Research Stambulian (CEIC), and by the Food and Drugs National Regulatory Agency (Administración Nacional de Medicamentos, Alimentos y Tecnología Médica, ANMAT), on March 22, 2022. Participants received compensation for their participation by travel costs reimbursement and food during their stay at the facilities.

Samples of the surveillance strategy implemented by the Ministry of Health of the Province of Buenos Aires that were used, were obtained from individuals that gave a written informed consent, after receiving a fully explanation of the nature and possible consequences of the study. The study was approved on February 3, 2022, by the Central Ethics Committee of Buenos Aires Province (Comité de ética central de la provincia de Buenos Aires).

## Vaccine
The vaccine was manufactured by Laboratorio Pablo Cassará S.R.L, according to good manufacturing practice guidelines. The recombinant protein was produced in a CHO-S-cell line and consisted of a single-chain dimer of the receptor binding domain (RBD), comprising amino acids 319 R to 537 K of the Spike protein from Gamma SARS-CoV-2 virus variant. ARVAC CG consisted in a liquid formulation containing 25 μg or 50 μg per 0.5 mL in a vial, with aluminum hydroxide as the adjuvant[13] (see Supplementary methods for more detailed description of vaccine manufacture and quality control).

## SARS-CoV-2 Neutralization assays
Neutralization assays were performed using live SARS-CoV-2 virus isolates[30]. Serial dilutions of plasma samples from 1/8 to 1/16384 were incubated 1 h at 37 °C in the presence of Ancestral (B.1), Gamma, Delta, or Omicron variants (BA.1 or BA.5) in Dulbecco's Modified Eagle Medium (DMEM), 2% fetal bovine serum (FBS). Then, 50 μL of the mixture were deposited over Vero cell monolayers for an hour at 37 °C (MOI, 0.01). Infectious medium was removed and replaced by DMEM, 2%-FBS. After 72 h, cells were fixed with 4% paraformaldehyde (4 °C, 20 min) and stained with crystal violet solution in methanol. The cytopathic effect (CPE) on the cell monolayer was assessed visually. If damage to the monolayer was observed in the well, it was considered manifestation of CPE and the neutralization titer was defined as the highest serum dilution that prevented any CPE. The neutralization antibody titers below the detection limit (1:8 dilution) were set as 4.

The nAb titers were transformed to international units per ml (IU/ml) by the inclusion in each plate of a secondary standard that was calibrated with the WHO international standard (NIBSC code: 20/268) following the WHO procedures manual[31].

## SARS-CoV-2 antibody ELISA
Antibodies against SARS-CoV-2 spike protein or RBD were detected using established, commercially available, two-step ELISAs COVIDAR[28], or SARS-CoV-2 (RBD) total Ab ELISA from DRG International (DRG Inc, Springfield, NJ 07081 USA) following manufacturer instructions. Data were collected using a Multiscan Go microplate reader from ThermoScientific with Thermo Scientific SkanIt Software. The immunoglobulin G (IgG) concentration of each sample, expressed in Binding Antibody Units/mL (BAU/mL) was calculated by extrapolation of the optical density at 450 nm (OD450) on a calibration curve built using serial dilutions of the WHO International Standard for anti-SARS-CoV-2 immunoglobulin.

## ELISpot
The T-cell mediated immune response against the SARS-CoV-2 RBD was assessed after the in vitro peptide stimulation of peripheral blood

mononuclear cells (PBMC), followed by IFN-γ and IL-4 enzyme-linked immune absorbent spot (IFN-γ (BD Biosciences) and IL-4 (Mab-Tech) ELISpot). A peptide pool of overlapping SARS-CoV-2 peptides, encompassing the SARS-CoV-2 spike RBD covering the Gamma variant was used in the assay (JPT Peptide Technologies GmbH, Germany). ElisPot Plates were scanned on an ImmunoSpot reader (Cellular Technology Ltd.). Specific spots were counted using the ImmunoSpot 5.0 software.

## Statistical analysis

Variables are reported as means +/− standard deviations, medians and interquartile range (IQRs) or CI95%, and numbers and percentages. Immunogenicity response was assessed by means of nAb geometric mean titers (GMTs), and seroconversion rates. Fourfold seroconversion (4×-seroconversion) or tenfold seroconversion (10×-seroconversion) were defined respectively as an increase in neutralizing antibodies equal or higher than four- or ten-times when the baseline nAb titers before the booster vaccine were detectable or four times the lower detection limit when the baseline concentration was not detectable. Differences in mean, geometric mean, or percentage values between groups and among prior vaccine platforms were assessed by means of Mann Whitney u test, Wilcoxon pair-matched test, Kruskal-Wallis test with post hoc Dunn's for multiple comparisons, Fisher exact test, or Chi-square distribution, as appropriate. 95% CI were calculated using the Wilson/Brown method. Missing data were assumed to be missing at random. Statistical analyses were done using GraphPad Prism v8.4.2 (GraphPad Software, San Diego, CA), Two-sided $P$-values < 0.05 were considered statistically significant.

## Reporting summary

Further information on research design is available in the Nature Portfolio Reporting Summary linked to this article.

## Data availability

All data are presented in the main text or the supplementary materials and are available upon reasonable request from the corresponding author. Data to be shared include all the individual participant data collected during the trial after deidentification and the study protocol. Data will be shared after completion of the trial and after publication of the results. Source data are provided with this paper.

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

## Acknowledgements

We thank the trial volunteers for their contribution and commitment to vaccine research. We are thankful to the staff of Laboratorio Pablo Cassará for their essential contribution to the development, scaling, manufacture, control and stability studies of the clinical batches of ARVAC CG antigen and vaccine: Krum, Valeria; Drehe, Ignacio; Zurvarra, Francisco; Baque, Jonathan; Payes, Cristian; Heinrich, Brenda; Gambone, Melisa; Descoins, Horacio; De Nichilo, Analía; Sidabra, Johanna; Licausi, Mariana; Cortez, Christian; Roman, María Victoria; Villar, Alejandra; Diaz, Sebastián; Frattolillo, Matías; Mestre, Diego; Gonzalo, Javier; Maluvini, Ernesto; Sperandini, Cecilia; Farre, Paola; Trovato, Fernando; Strada, Ariel; Cocciolo, Giovanna; and Privitera, Anabella. We additionally thank Roberto Debagg for giving valuable advice during to the design and completion of the clinical trial. We thank Daniela Hozbor for the access to the samples obtained during the COVID-19 serology surveillance strategy implemented by the Ministry of Health of the Province of Buenos Aires. We thank Florencia Quiroga, Gabriela Turk and Natalia Laufer for performing T cell stimulation assays by ELISpot at Instituto de Investigaciones Biomédicas en Retrovirus y SIDA, INBIRS-CONICET, Buenos Aires, Argentina. We also thank Jaime Torres who provided medical writing support. We thank Elisa Estenssoro of the Ministry of Health of the Province of Buenos Aires for critical review of the manuscript. We are grateful to Dr. Ángela Spanguolo de Gentile, Dr. Pablo Bonvehí, and Dr. Hugo Krupitzki who are members of the external Independent Committee of Data Review for their dedication and continuous invaluable advice. The study was funded by Laboratorio Pablo Cassará S.R.L. (Argentina). Laboratorio Pablo Cassará, in collaboration with the other authors, had a role in the design and supervision of the study.

## Author contributions

J.C., M.L., J.C.V., F.M.O., J.M.R., J.F., K.A.P., and L.M.C. conceptualized and designed the study. K.A.P., L.M.C., and J.C. wrote and edited the manuscript. M.L., F.M.O., and J.M.R. were responsible for project administration. J.C., K.A.P., L.M.C., M.L., and J.F., accessed, verified, and analyzed the data, and generated the interim report. E.F. is the study site scientific director and supervised the study. G.A.Y. is a study site principal investigator. K.H., G.A.Y., and P.E.P.L. were responsible for the site work including the recruitment, follow up, and data collection. B.M., M.S., A.D., C.P.C., L.S., L.B. performed immunogenicity experiments. A.C., A.V., D.A., and J.G. contributed to the performance and analysis of neutralizing antibody assays. S.G., and VVGM provided the plasma samples from individuals boosted with authorized vaccines. J.M.R., J.C.V., F.M.O. and L.P.C. R&D and C.M.C. consortium were responsible for R&D for chemistry, manufacture and controls of antigen and study products. J.C.V. and F.M.O. provided regulatory oversight. ML was responsible for the overall supervision of the study and monitored the trial. All authors contributed to data interpretation, review, and editing of this manuscript. All authors have read and approved the final version of the manuscript.

## Competing interests

J.M.R., L.P.C. R&D and CMC Group, F.M.O., J.C.V. are employees of Laboratorio Pablo Cassará S.R.L., which developed the vaccine and funded the trial. M.L. and J.F. are external consultants and received honoraria from Laboratorio Pablo Cassará S.R.L. All other authors declare no competing interests.

## Additional information

[1]Instituto de Investigaciones Biotecnológicas, Universidad Nacional de San Martín (UNSAM) – Consejo Nacional de Investigaciones Científicas y Técnicas (CONICET), San Martín (1650), Buenos Aires, Argentina. [2]Escuela de Bio y Nanotecnologías (EByN), Universidad Nacional de San Martín, San Martín (1650), Buenos Aires, Argentina. [3]Instituto de Investigaciones Biomédicas en Retrovirus y SIDA, INBIRS-CONICET, Facultad de Medicina UBA, Buenos Aires, Argentina. [4]Fundación Pablo Cassará - Unidad de I + D de Biofármacos, Saladillo 2452 C1440FFX, Ciudad Autónoma de Buenos Aires, Argentina. [5]Ministerio de Salud de la Provincia de Buenos Aires, Buenos Aires, Argentina. [6]FP CLINICAL PHARMA, Ciudad Autónoma de Buenos Aires, Buenos Aires, Argentina. [7]Laboratorio Pablo Cassará – Unidad de I + D de Biofármacos, Saladillo 2452 C1440FFX, Ciudad Autónoma de Buenos Aires, Argentina. [8]Nobeltri, Ciudad Autónoma de Buenos Aires, Argentina. ✉e-mail: kpasquevich@iib.unsam.edu.ar; jucassataro@iib.unsam.edu.ar

## Laboratorio Pablo Cassará R&D and CMC for ARVAC CG consortium

Sabrina A. del Priore[7], Andrés C. Hernando Insua[1,4], Ingrid G. Kaufmann[7], Adrián Di María[1], Adrián Góngora[7], Agustín Moreno[7], Susana Cervellini[7], Blasco Martin[7], Esteban Ali[7], Romina Albarracín[7], Barsanti Bruno[7], Fernando Toneguzzo[7], Guillermina Sasso[7], Sebastian Stamer[7], Regina Cardoso[7] & Alejandro Chajet[7]

