## [Peer Review File · Nature Communications]

Safety and immunogenicity of a SARS-CoV-2 Gamma variant RBD-based protein adjuvanted vaccine used as booster in healthy adults.Reviewers' Comments:

Reviewer #1:

Remarks to the Author:

What are the noteworthy results?

The significance of the work is moderate

Will the work be of significance to the field and related fields? How does it compare to the established literature? If the work is not original, please provide relevant references.

Several Covid-19 vaccine are base also on the recombinant RBD protein expressed in several host. For example pichia and particularly in CHO cells. The tandem repeat dimer or the naturally occurring dimer were already used as vaccine with a large experience. This is the first time the VOC RBD gamma is used however comparing with previous work the result of this clinical trial didnt provide enough new information

Does the work support the conclusions and claims, or is additional evidence needed?

the paper is confused as people were vaccinated with two doses and this was claim as booster dose. Are there any flaws in the data analysis, interpretation and conclusions? Do these prohibit publication or require revision?

No

Is the methodology sound? Does the work meet the expected standards in your field?

The work meet the standard of a clinical trial phase I, but there are plenty of clinical trial as primary schedule or as a booster

Is there enough detail provided in the methods for the work to be reproduced?

Yes but the contribution isnot really substantial to the field. I recomend to be rejected and encourage author to submit to a more conventional journal

Reviewer #2:

Remarks to the Author:

The work described in the manuscript submitted by Pasquevich et al is a straightforward Phase 1 dose-acceleration study that contains interesting observations about a novel recombinant protein+adjuvant (aluminum hydroxide) vaccine. The vaccine was produced in a CHO-S-cell line consisting of a single-chain dimer of the receptor binding domain (RBD: 319R to 537K aa) of the Spike protein of a Gamma variant. The authors delivered two doses of either 25µg (target N=60) or 50 µg (target N=20) intramuscularly 28 days apart and reported AEs aftet each dose (solicited up to 28 days post-dose). Immune responses were assessed using what appears to be a standard 'in-house' end-point dilution, live-virus plaque reduction neutralization assay presumably perfomed in Dr Geffner's laboratory (neutralizing antibody (Nab) responses), commercially available ELISA kits ("COVIDAR or SARS-CoV-2 (RBD) total Ab ELISA from DRG International") and ELISpots for IFN-γ and IL-4 using a Gamma peptide pool for ex vivo re-stimulation. Neutralizing antibody responses were assessed against a relevant panel of live viruses including ancestral, Gamma, Delta and Omicron (BA.1 and BA.5) and the authors include an international standard so their results can be more easily compared with what others have reported (NIBSC code: 20/268). The subjects recruited had received a range of different vaccines for their primary series (most Oxford/AZ, Sputnik or Sinopharm) so this was a 'double' heterologous booster study: both the vaccine itself and the targeted antigen in the booster were different from primary series. Although others have reported responses to recombinant spike vaccines containing multiple 'hot spot' mutations (some of which are found in the Gamma variant (eg: Wang Z et al npj Vaccines 2022), I'm not aware of any other report of a 'pure' Gamma-based vaccine. As a result, this work is quite novel. The vaccine appears to have been well-tolerated as would be expected for an alum-adsjuvanted recombinant formulation and highly immunogenic (with some caveats – see concerns below).

Major Concerns

The study was designed to test this vaccine at two doses levels and after two doses 28 days apart and the manuscript contains safety data out to 28 days after each dose. Unfortunately, the immune outcomes are only reported to day 14 or 28 after the first dose. Although these data are actually quite promising for the use of a single dose of this vaccine as a booster (ie: strong NAb at 14 and 28 days against the range of variants tested as well as decent IFN- γ ELISpot responses at 28 days)(Figures 3,4 and 6) regardless of which vaccine was used in the primary series ... the report feels incomplete without the immune outcomes after the second dose. This is particularly true since the NAb data presented in Figure 5 where both D14 and D28 results are shown suggest that the antibody response to the first booster dose may be very transient. In Figure 5, the NAb titres against many of the variants tested actually fall substantially between D14 and D28 in 7 of the 10 cases (2 doses levels, 5 variants tested). The Gamma NAb response – for example – falls from 844 \rightarrow 322 between D14 and D28. Such a rapid fall in titres is unexpected and quite worrisome if this vaccine will be used as a 'single dose booster'. Unfortunately, the study was not designed to assess longer-term responses to a single dose. This observations should – at the very least – be addressed in the Discussion. Even better, the authors should be asked to provide their data for the immune outcomes after the second dose.

It is difficult to compare the responses to the 25 and 50 μ g doses because almost all of the higher dose group received the Sputnik vaccine as their primary series while only \sim 1/3 of the subjects who received the lower dose got Sputnik initially. This difference may help to explain some of the odd findings between the high and low dose groups (eg: the generally lower responses in the high dose group who were seropositive at baseline (Supplementary Figure 1). Since there were approximately equal numbers of Spuntik recipients in the high and low dose groups, a Supplementary Figure focused on these subjects alone might be useful.

The fact that almost 100% of the subjects in the higher dose are of the study received Sputnik as their primary series needs to be highlighted as a limitation. Comparisons across groups when the groups are so different is problematic.

Since this appears to be a first report, it would be very helpful to have a more detailed Supplementary Methods section providing some characterization of this new vaccine (ie: CMC information). For example, the only thing we are told about the vaccine can be found between lines 294-298: "a CHO-S-cell line and consisted of a single-chain dimer of the receptor binding domain (RBD), comprising amino acids 319R to 537K of the Spike protein from Gamma SARS-CoV-2 virus variant. ARVAC CG consisted in a liquid formulation containing 25- μ g or 50- μ g per 0.5 mL in a vial, with aluminum hydroxide as the adjuvant¹³" We are not told how much alum was used. Although we are told that each vial of vaccine contains 0.5mL, we are not told what the actual IM injection volume is ...

Minor Concerns

Figure 4

The use of solid bars in Figure 4 is misleading since many of the 'bars' represent data from a single subject.

Table 2 The use of the convenience samples from another BT/Pf vaccine booster study older is also problematic since the subjects in this study were quite different from the Gamma vaccine recipients in several ways (eg: 36.5 versus 32 and 27yo in groups A and B respectively) and 100% seronegative at baseline versus 92 and 80% respectively. The time since the primary series also quite different (4.2 months in the BT/Pf subjects versus 7.9 and 6.6 months respectively). These differences should be explained more clearly and included among the limitations of the study.

Line 52-53 the authors state "These virus variants are, in general, more contagious, and virulent than

previous strains". While there is no doubt that some of the newer variants have been more contagious, there is very little evidence they are more virulent ...

Although the text is quite easy to flow, there are minor issues throughout (ie: missing words, grammar, etc) that would benefit from a thorough editorial review. For example:
Line 65-66 ". Therefore, pandemic remains threat unless most people get vaccinated."

Line 294 "... recombined protein ..."→ "should be "recombinant protein"

Reviewer #3:

Remarks to the Author:

In this paper safety and immunologic results were presented from 2 cohorts of participants, those that received two different doses of a booster of ARVAC CG following a different COVID-19 vaccine platform. They also presented immunologic results from a cohort that received a booster of an mRNA vaccine. They examine the neutralizing antibody response against the SARS-CoV-2 Ancestral strain and several variants of concern (Gamma, Delta, Omicron BA.1 and Omicron BA.5) measured by a live virus-based neutralization assay

Questions/comments:

- 1) In the abstract the authors do not mention the cohort of participants that they report on who received a mRNA vaccine. Since the manuscript treats the results from this cohort on a similar level as the cohorts receiving the ARVAC CG vaccine this should be discussed in the abstract.
- 2) Line 54. The sentence says virus variants are in general more contagious and virulent than previous strains. I do not believe the variants have been shown to be more virulent (in fact the opposite may be true). The references given do not support the statement of virulence.
- 3) Line 116. Does 4x mean 4 fold? This is not standard notation. If you are going to use this notation you must define it.
- 4) Line 187. A per protocol population was mentioned but not defined. Please describe the populations analyzed.
- 5) Line 259 says participants were sequentially assigned to one of two groups but does not say why 3 times as many participants received the lower dose and how it was determined which dose a participant received.
- 6) The protocol does not adequately describe missing data. Line 266 says missing data was infrequent and inconsequential. This is not adequate. for each time point with antibody results we need to know if there were missing data. So provide an n in each figure (if less than at baseline). If no missing antibody results then this can be stated once at the beginning.
- 7) Line 331 states "Nonparametric statistical tests were used to analyze titer data" but I did not seem the used for categorical variables. I think this sentence can be deleted since there is a better description later in the paragraph.
- 8) Time since infection is quite different between two dose groups. This probably should be mentioned. Also, the percentage with previous covid seems to be quite a bit larger for the higher dose level. Similarly the mRNA cohort has no participants with prior COVID-19. Please add these issues to the discussion.

Point-by-point response to the reviewers' comments

We would like to thank the editor and the reviewers for their valuable comments. We have carefully revised our manuscript according to the comments and suggestions of the reviewers. The comments were very helpful for revising our paper.

Please find below the point-by-point response to the reviewers' comments. Reviewers' comments are written in black. Responses in blue.

REVIEWER COMMENTS

Reviewer #1 (Remarks to the Author):

What are the noteworthy results? The significance of the work is moderate.

Will the work be of significance to the field and related fields? How does it compare to the established literature? If the work is not original, please provide relevant references. Several Covid-19 vaccines are based also on the recombinant RBD protein expressed in several host. For example picchia pastoris and particularly in CHO cells. The tandem repeat dimer or the naturally occurring dimer were already used as vaccine with a large experience. This is the first time the VOC RBD gamma is used however comparing with previous work the result of this clinical trial did not provide enough new information

Does the work support the conclusions and claims, or is additional evidence needed? the paper is confused as people were vaccinated with two doses and this was claim as booster dose.
Are there any flaws in the data analysis, interpretation and conclusions? Do these prohibit publication or require revision? No

Is the methodology sound? Does the work meet the expected standards in your field? The work meet the standard of a clinical trial phase I, but there are plenty of clinical trial as primary schedule or as a booster

Is there enough detail provided in the methods for the work to be reproduced? Yes but the contribution is not really substantial to the field. I recommend to be rejected and encourage author to submit to a more conventional journal

Replay: We appreciate that the reviewer acknowledges that there are no problems with data analysis, interpretation and conclusion and no additional evidence is needed. Also, that the work meets the expected standards in the field and that the methods are provided with enough details.

Regarding the contribution to the field, to our knowledge, this is the first clinical trial reported evaluating a Gamma variant RBD protein-adjuvanted vaccine, used as heterologous booster of different primary series vaccine platforms. In contrast with other phase 1 studies, in this study the subjects received a range of different vaccines for their primary series (adenoviral, inactivated, heterologous), which allows to evaluate the performance of the recombinant vaccine as booster of different vaccine platforms. Moreover, as stated by reviewer 2, this is a 'double' heterologous booster study: both the vaccine itself and the targeted antigen in the booster were different from primary series.

The results of the Phase 1 study presented in this manuscript contain not only the safety data about the vaccine (primary endpoint of the study) but also many immunogenicity data, including determination of nAb titers against several VOCs of SARS-CoV-2 and evaluation of cellular immune responses.

In addition, this is the first Covid-19 vaccine that has been developed from bench to the clinic by local researchers and a local pharmaceutical company in South America. Therefore, it represents a key project regarding health and scientific advancement at national level, and a significant achievement for the research

and medical communities in our region. Besides, this development has implications regarding vaccine availability and supply for our region.

Reviewer #2 (Remarks to the Author):

The work described in the manuscript submitted by Pasquevich et al is a straightforward Phase 1 dose-acceleration study that contains interesting observations about a novel recombinant protein+adjuvant (aluminum hydroxide) vaccine. The vaccine was produced in a CHO-S-cell line consisting of a single-chain dimer of the receptor binding domain (RBD: 319R to 537K aa) of the Spike protein of a Gamma variant. The authors delivered two doses of either 25µg (target N=60) or 50 µg (target N=20) intramuscularly 28 days apart and reported AEs after each dose (solicited up to 28 days post-dose). Immune responses were assessed using what appears to be a standard 'in-house' end-point dilution, live-virus plaque reduction neutralization assay presumably performed in Dr Geffner's laboratory (neutralizing antibody (Nab) responses), commercially available ELISA kits ("COVIDAR or SARS-CoV-2 (RBD) total Ab ELISA from DRG International") and ELISpots for IFN-γ and IL-4 using a Gamma peptide pool for ex vivo re-stimulation. Neutralizing antibody responses were assessed against a relevant panel of live viruses including ancestral, Gamma, Delta and Omicron (BA.1 and BA.5) and the authors include an international standard so their results can be more easily compared with what others have reported (NIBSC code: 20/268). The subjects recruited had received a range of different vaccines for their primary series (most Oxford/AZ, Sputnik or Sinopharm) so this was a 'double' heterologous booster study: both the vaccine itself and the targeted antigen in the booster were different from primary series. Although others have reported responses to recombinant spike vaccines containing multiple 'hot spot' mutations (some of which are found in the Gamma variant (eg: Wang Z et al npj Vaccines 2022), I'm not aware of any other report of a 'pure' Gamma-based vaccine. As a result, this work is quite novel. The vaccine appears to have been well-tolerated as would be expected for an alum-adsorbed recombinant formulation and highly immunogenic (with some caveats – see concerns below).

We thank the reviewer for the valuable comments, which in our opinion have further improved our manuscript.

Major Concerns

1- The study was designed to test this vaccine at two doses levels and after two doses 28 days apart and the manuscript contains safety data out to 28 days after each dose. Unfortunately, the immune outcomes are only reported to day 14 or 28 after the first dose. Although these data are actually quite promising for the use of a single dose of this vaccine as a booster (ie: strong NAb at 14 and 28 days against the range of variants tested as well as decent IFN-γ ELISpot responses at 28 days) (Figures 3,4 and 6) regardless of which vaccine was used in the primary series ... the report feels incomplete without the immune outcomes after the second dose. This is particularly true since the NAb data presented in Figure 5 where both D14 and D28 results are shown suggest that the antibody response to the first booster dose may be very transient. In Figure 5, the NAb titres against many of the variants tested actually fall substantially between D14 and D28 in 7 of the 10 cases (2 doses levels, 5 variants tested). The Gamma NAb response – for example – falls from 844 → 322 between D14 and D28. Such a rapid fall in titres is unexpected and quite worrisome if this vaccine will be used as a 'single dose booster'.

Replay: We agree with the reviewer on the importance of including the data on immune responses after the second booster, and accordingly, we analyzed and included such information in the reviewed version of the manuscript (see more explanation on this in point 2).

Regarding the falling off in antibody titers that the reviewer indicated, it must be emphasized that similar kinetics in nAb titers, reaching a maximum level shortly after vaccination and then slowly declining, have been shown as well for other vaccines used as primary or booster immunization, although differences in the decay rates between vaccines have been shown ^{1,2}. For example, Canetti et. al ³ described that after a fourth dose with the mRNA1273 COVID19 vaccine the nAb titers against the Wuhan strain reached a peak after 14 days of booster (GMT 5,580), which declined to a GMT of 3,064 only one week later (45%

reduction in nAb GMT). Whereas the same study showed that after a fourth dose with BNT16b2 the nAb titers reached a peak 2-3 weeks later, that slowly declined to basal levels after 3 months of booster.

2- Unfortunately, the study was not designed to assess longer-term responses to a single dose. These observations should – at the very least – be addressed in the Discussion. Even better, the authors should be asked to provide their data for the immune outcomes after the second dose.

Replay: Since this was a first in human study, safety after one and two vaccine doses was the main endpoint. Our findings highlight the excellent safety profile of the vaccine. These results were key to allow for further clinical development of this vaccine and made possible the initiation of an ongoing phase 2/3 study in which a single administration of the vaccine as a booster dose is being evaluated. In this new study, immunogenicity is the primary endpoint and longer time points after vaccine administration are tested, together with a larger population, including older adults and volunteers with and without comorbidities.

Neutralizing antibody (nAb) responses after the 2nd vaccine administration (separated 28 days from first booster) were analyzed recently and therefore were not included in the first version of the manuscript. In accordance with the reviewer's suggestion, we now added results of the immune response after the second administration of the vaccine (Supplementary Fig. 10). After the second ARVAC CG dose the nAb titers remained significantly higher than those at baseline, but no booster effect was observed in comparison with the first dose. The lack of booster effect may be due to the short interval between doses that may not be the optimal in terms of immunological performance. It is important to state that this short dosing interval and the application of a second booster in the study protocol were chosen to collect safety data in a short time period that would allow the study of the vaccine in a primary two dose regimen rather than to evaluate the immunogenicity after a second dose.

We added the following Sentences in the results section (Page 11, lines 202-204):

“A second booster with ARVAC CG was given to the volunteers after 28 days to collect safety data after two vaccine administrations. The nAb titers remained significantly higher than baseline values (1d) after 42 days and 56 days of first dose administration (Supplementary Fig.10)”

We added following Sentences in the discussion section (Page 14, lines 282-287):

“Immune responses after a single booster dose presented were could be assessed only after 14 and 28 days of one booster administration. After the second ARVAC CG administration the nAb remained significantly higher than baseline but no booster effect was observed. The lack of booster effect may be due to the short interval between boosters, that may not be the optimal in terms of immunological performance 24-26. Longer-term follow-up of immune responses after a single booster dose will have to be studied in an ongoing phase 2/3 study.”

3- It is difficult to compare the responses to the 25 and 50µg doses because almost all of the higher dose group received the Sputnik vaccine as their primary series while only ~1/3 of the subjects who received the lower dose got Sputnik initially. This difference may help to explain some of the odd findings between the high and low dose groups (eg: the generally lower responses in the high dose group who were seropositive at baseline (Supplementary Figure 1). Since there were approximately equal numbers of Spuntik recipients in the high and low dose groups, a Supplementary Figure focused on these subjects alone might be useful.

Replay: We appreciate the reviewer's suggestion to clarify the comparison between both groups. In such regard, when the 60 participants in group A were stratified by their primary vaccination schedule, there were 20 participants that received the inactivated BBIBP-CorV vaccine, 17 participants that received the ChadOx1-S vaccine, 21 participants that received rAd26/rAd5 (Sputnik V) vaccine, whereas only one participant received an heterologous regimen (rAd26 and ChadOx1-S), and another participant received the single dose primary regimen of Ad5-nCov (Cansino) vaccine. In group B, of the 20 enrolled participants, 14 participants had received the inactivated BBIBP-CorV vaccine, 1 participant had received the ChadOx1-S

vaccine, 1 participant had received rAd26/rAd5 (Sputnik V) vaccine, 3 participants received an heterologous regimen, and another participant received the single dose primary regimen of Ad26.Cov2.S (Janssen) vaccine. Thus, we believe that when the reviewer says Sputnik it could be BBIBP-CorV vaccine?

Since there were approximately equal numbers of BBIBP-CorV recipients in the high and low dose groups, we re-analyzed the data of supplementary Figure 1, as suggested by the reviewer, focusing on the BBIBP-CorV vaccinated subpopulation and made a new supplementary figure. Results indicate that in the subgroup of subjects who received BBIBP-CorV as primary vaccination, the GMFR in nAb titers from baseline were similar between anti-N seronegative or anti-N seropositive individuals whether in group A or B (Supplementary Fig. 2).

We added this new Figure to the supplementary material of the manuscript (supplementary Figure 2) and a sentence in the results section of the manuscript (addition is denoted in underlined text) (Page 8, lines 144-146):

“Moreover, GMFR from baseline were similar for both populations in groups A and B either when analyzed in all individuals of each group (Supplementary Fig. 1) or in the subgroup of subjects who received BBIBP-CorV as primary vaccination (Supplementary Fig. 2)”.

The fact that almost 100% of the subjects in the higher dose are of the study received Sputnik as their primary series needs to be highlighted as a limitation. Comparisons across groups when the groups are so different is problematic.

Replay: Again, we believe that the reviewer refers to the subjects who received BBIBP-CorV as primary vaccination series. To answer the reviewer’s concern, we reanalyzed the differences in nAbs between both dosage groups in the subgroup of individuals that had received the BBIBP-CorV vaccine as primary vaccination scheme (n=20 in group A, and n=14 in group B). Of note, we found similar differences between both dosage groups than those observed when all participants were considered as a whole. The GMFR in nAb titers against Ancestral, Gamma, Omicron BA.1 and Omicron BA.5 VOCs were significantly higher in Group B than in Group A (P=0.0154, P=0.0118, P=0.0160, and P=0.0459, respectively). Fourteen days after boosting, 4×-seroconversion rates for all tested variants were similar in both groups, whereas the 10×-seroconversion rates for the Ancestral and Omicron BA.1 VOC were significantly higher in the 50-µg cohort than in the 25- µg cohort.

These results were added to the results section of the manuscript, Page 9 lines 154-161, Supplementary Table 3.

“Comparisons between group A and B in terms of GMFR of nAb titers and seroconversion rates was assessed in the subgroup of individuals that had received the BBIBP-CorV vaccine as primary vaccination scheme. The GMFR in nAb titers against Ancestral, Gamma, Omicron BA.1 and Omicron BA.5 VOCs were significantly higher in Group B than in Group A (P=0.0154, P=0.0118, P=0.0160, and P=0.0459, respectively). Fourteen days after boosting, 4×-seroconversion rates for all tested variants were similar in both groups, whereas the 10×-seroconversion rates for the Ancestral and Omicron BA.1 VOC were significantly higher in the 50 µg cohort than in the 25 µg cohort (Supplementary Table 3).”

And in the discussion section Page 12, lines 221-229:

“The differences in the proportions of subjects with different types of primary vaccine regimen in the low-dose and the high-dose groups may difficult comparisons. However, the possibility to include different primary vaccination schemes was important in this phase 1 study to have a representation of the diversity in primary vaccination schemes that were used in Argentina. Since most individuals in group B had received the BBIBP-CorV vaccine as primary vaccination regimen and there were approximately equal numbers of BBIBP-CorV recipients in the high and low dose groups, the comparison of these subgroups was also performed. Similar to the findings when all volunteers were included, in these more homogeneous subpopulations, the 50 µg dose was consistently more immunogenic than the 25 µg dose.”

4- Since this appears to be a first report, it would be very helpful to have a more detailed Supplementary Methods section providing some characterization of this new vaccine (ie: CMC information). For example, the only thing we are told about the vaccine can be found between lines 294-298: “a CHO-S-cell line and consisted of a single-chain dimer of the receptor binding domain (RBD), comprising amino acids 319R to 537K of the Spike protein from Gamma SARS-CoV-2 virus variant. ARVAC CG consisted in a liquid formulation containing 25-µg or 50-µg per 0.5 mL in a vial, with aluminum hydroxide as the adjuvant¹³” We are not told how much alum was used. Although we are told that each vial of vaccine contains 0.5mL, we are not told what the actual IM injection volume is ...

Replay: following the reviewer’s suggestion, we added a more detailed supplementary Methods section providing the characterization of the vaccine. See revised supplementary material (pages 24-26, lines 207-290), supplementary Fig. 11 and supplementary Fig. 12.

Minor Concerns

5- Figure 4. The use of solid bars in Figure 4 is misleading since many of the ‘bars’ represent data from a single subject.

Replay: The original Figure 4 was a combination of a scatter plot (where each point represents a subject) and bars showing the geometric mean and 95% CI. In those subgroups with only a volunteer the point and the geometric mean were coincident, and no 95% CI was shown. The number of individuals in each subgroup was also shown at the bottom of each bar. Since there are so many data, we found that this representation was the best way to visualize the results, with different colors per bar for each vaccine regimen.

To attend the reviewer’s concern, we changed figure 4 showing scatter plots without the bars. You will find the new Figure 4 in the revised manuscript.

6- Table 2. The use of the convenience samples from another BT/Pf vaccine booster study older is also problematic since the subjects in this study were quite different from the Gamma vaccine recipients in several ways (eg: 36.5 versus 32 and 27 yo in groups A and B respectively) and 100% seronegative at baseline versus 92 and 80% respectively. The time since the primary series also quite different (4.2 months in the BT/Pf subjects versus 7.9 and 6.6 months respectively). These differences should be explained more clearly and included among the limitations of the study.

Replay: Regarding the mRNA cohort, while all the subjects of this cohort declared no previous infection, serological analysis of anti-N antibodies indicated that half of them may have had previous contact with the SARS-Cov-2 virus. This proportion is similar to that observed in group A and group B. Regarding the differences in time since primary vaccination to booster, to address the reviewer concern we compared the booster effect of the BNT16b2 with the booster effect in the ARVAC CG in subjects with short time since primary vaccination series completion and obtained similar results as those with the whole population (supplementary Fig. S6). In addition, since 16 of the 18 BNT16b2 boosted subjects had a rAd26/rAd5 (Sputnik V) vaccination scheme, we added a new figure with the nAb responses in individuals Sputnik V as primary vaccination scheme. Similar results were obtained when analyzing these more homogenous subpopulations. As suggested by the reviewer, we added these results (supplementary Fig 7) and discussed these limitations in the discussion of the manuscript:

Results section, Page 10, Lines 179-183:

“Similar results were obtained when the nAb responses of the BNT16b2 group were compared to those of ARVAC CG boosted individuals whose time since completion of primary vaccination series and booster was less than 180 days (Supplementary Fig. 6) or when only the individuals whose primary vaccination scheme was rAd26/rAd5 (Sputnik V vaccine) were analyzed (Supplementary Fig. 7).”

Discussion section, Pages 13-14, lines 264-279:

“The comparison of the immunogenicity results of ARVAC CG cohorts with those of a contemporaneous BNT16b2 booster study has another limitation, since the study participants presented some demographic differences in their ages (36.5 in BNT16b2 group versus 32 and 27 years in groups A and B, respectively). Although the time from last SARS-CoV-2 vaccine dose (4.2 months in the BNT16b2 boosted subjects versus 7.9 and 6.6 months in group A and B, respectively) was different this did not influence the immune outcomes in this study. Peak of nAb titers reached after booster in both ARVAC CG cohorts were similar whether the time since primary vaccination completion to booster was short (<180 days) or long (≥180 days). Indeed, ARVAC CG boosters at short time (<180 days) showed a better performance than BNT16b2 booster. The confirmed previous SARS-CoV-2 infection history (0% in group BNT16b2 versus 8 and 20% in groups A and B, respectively) was quite different, nevertheless anti-N antibodies serology indicated that the three populations had similar proportions of seropositive individuals and might had similar previous contact with the virus. The proportion of primary vaccination schemes was also different, since most individuals in BNT16b2 boosted cohort had received the rAd26/rAd5 (Sputnik V) vaccine as primary vaccination regimen, however the comparison among subjects with the same primary vaccination led to similar results.”

7- Line 52-53 the authors state “These virus variants are, in general, more contagious, and virulent than previous strains”. While there is no doubt that some of the newer variants have been more contagious, there is very little evidence they are more virulent ...

Replay: We thank the reviewer for pointing out this mistake, we deleted the statement mentioning that the virus variants are more virulent than previous strains. Page 5, line 62.

8- Although the text is quite easy to flow, there are minor issues throughout (ie: missing words, grammar, etc) that would benefit from a thorough editorial review. For example: Line 65-66 “. Therefore, pandemic remains threat unless most people get vaccinated.”

Line 294 “... recombined protein ...”→ ‘should be “recombinant protein”

Replay: We thank the reviewer for pointing out these mistakes. The text was modified accordingly.

Page 5, Line 75: Therefore, pandemic remains a threat unless most people get vaccinated.

Page 18, Line 373: recombinant

Reviewer #3 (Remarks to the Author):

In this paper safety and immunologic results were presented from 2 cohorts of participants, those that received two different doses of a booster of ARVAC CG following a different COVID-19 vaccine platform. They also presented immunologic results from a cohort that received a booster of an mRNA vaccine. They examine the neutralizing antibody response against the SARS-CoV-2 Ancestral strain and several variants of concern (Gamma, Delta, Omicron BA.1 and Omicron BA.5) measured by a live virus-based neutralization assay.

We thank the reviewer for the valuable comments, which in our opinion have further improved our manuscript.

Questions/comments:

1) In the abstract the authors do not mention the cohort of participants that they report on who received a mRNA vaccine. Since the manuscript treats the results from this cohort on a similar level as the cohorts receiving the ARVAC CG vaccine this should be discussed in the abstract.

Replay: Following the reviewer’s suggestion, we have added in the abstract a mention to the cohort that received a mRNA vaccine and the results obtained. Page 3, lines 48-50:

“Samples from participants of a surveillance strategy implemented by the local Ministry of Health that were boosted with BNT16b2 were also analyzed to compare with the booster effect of ARVAC-CG.”

2) Line 54. The sentence says virus variants are in general more contagious and virulent than previous strains. I do not believe the variants have been shown to be more virulent (in fact the opposite may be true). The references given do not support the statement of virulence.

Replay: This mistake was also a concern from reviewer 2, we deleted that the virus variants are more virulent than previous strains. Page 5, line 62.

3) Line 116. Does 4x mean 4 fold? This is not standard notation. If you are going to use this notation you must define it.

Replay: Yes, it means 4-fold. The definition of this notation was in the table legend and is needed since we analyzed two different seroconversion rates (fourfold and tenfold). We agree with the reviewer that is better to define it in the text.

Thus, corresponding changes were made in the results section, Page 8, lines 127-129:

“Seroconversion rates were evaluated as the percentage of subjects with at least a fourfold increase (4×-seroconversion rates) or a tenfold increase (10×-seroconversion rates) in the nAb titers at a specific timepoint respect to baseline values.”

And in the methods section, Page 20, lines 409-413:

“Fourfold seroconversion (4×-seroconversion) or tenfold seroconversion (10×-seroconversion) were defined respectively as an increase in neutralizing antibodies equal or higher than four- or ten-times when the baseline nAb titers before the booster vaccine were detectable or four times the lower detection limit when the baseline concentration was not detectable.”

4) Line 187. A per protocol population was mentioned but not defined. Please describe the populations analyzed.

Replay: The per protocol population included was described in the Methods section and in table 1. In the description it was missing that the participants must have a complete COVID-19 primary vaccination schedule; we added this in the revised manuscript.

In addition, it is possible that mentioning a per protocol population the reader may think that only a subpopulation was analyzed, therefore we changed the phrase mentioned by the reviewer.

Page 15-16, lines 309-318 of the revised manuscript.

“In this open-label, first-in-human, dose-escalation, phase 1 clinical trial, eligible volunteers were healthy men and nonpregnant women, aged 18 to 55, with a body-mass index of 18 to 30 with a complete COVID-19 vaccine primary schedule. Health status, assessed during the screening period, was based on medical history and extensive clinical laboratory tests, vital signs, and physical examination. Participants with a history of SARS-Cov-2 infection or COVID-19 within 60 days prior to recruitment into the study, or who tested positive in real-time polymerase-chain-reaction (RT-PCR) assay at screening or worked in an occupation with high risk of exposure to SARS-CoV-2, as well as those with an incomplete COVID-19 vaccine primary schedule, or who had received the last COVID19 primary vaccine shot within 4 months prior to recruitment into the study or have received a booster dose of any COVID-19 vaccine, were excluded.”

Page 11, line 218-220 of the revised manuscript:

“The booster effect after one dose of ARVAC CG vaccine was evident despite the variety of immunization schemes received by the study participants. ~~per protocol included population.~~”

5) Line 259 says participants were sequentially assigned to one of two groups but does not say why 3 times as many participants received the lower dose and how it was determined which dose a participant received.

Replay: The dose that each participant received was based on a sequential assignment plan. In the first stage of enrollment, the first five enrolled participants received the low dose vaccine (25 µg/dose). Only one participant per day was vaccinated. Afterwards in the second stage of enrollment, participants 6th to 10th received the high dose vaccine formulation (50 µg/dose). Only one participant per day was vaccinated. The next fifty-five enrolled participants received the 25 µg/dose (stage 3), and then the last fifteen participants received the 50 µg/dose (stage 4).

The low dose group (25 µg/dose) included 60 participants and the high dose group (50 µg/dose) included 20 participants. This number of participants allowed to obtain enough safety data for the low and the high vaccine dose and to increase the diversity regarding the primary vaccination scheme that were used in Argentina. The decision to include more participants in the low-dose group than in the high-dose group was based in the available literature from other RBD based vaccines, preclinical data, and pricing of the vaccine, which suggested that this dose would be safe and immunogenic enough; therefore, it was finally selected. However, based on the results of this trial and the decision to test bivalent vaccines, the high dose was finally selected to continue the clinical development.

We added the description of sequential assignment to the methods section of the protocol (Page 16, lines 324-330):

“A sequential assignment plan was prespecified in the study protocol. In the first stage of enrollment, the first five enrolled participants received the low dose vaccine (25 µg/dose). Only one participant per day was vaccinated. Afterwards in the second stage of enrollment, participants 6th to 10th received the high dose vaccine formulation (50 µg/dose). Only one participant per day was vaccinated. The next fifty-five enrolled participants received the 25 µg/dose (stage 3), and then the last fifteen participants received the 50 µg/dose (stage 4).”

6) The protocol does not adequately describe missing data. Line 266 says missing data was infrequent and inconsequential. This is not adequate. for each time point with antibody results we need to know if there were missing data. So, provide an n in each figure (if less than at baseline). If no missing antibody results, then this can be stated once at the beginning.

Replay: We thank the reviewer for pointing out this missing (or unclear) information. The sentence in line 266 was about the collection of safety data (local and systemic AEs as well as routine biochemical laboratory data).

Regarding antibody data at baseline and after 14 days of first dose, results are available for all study participants. Figures and tables including information at baseline and after 14 days of vaccination show the data of all study participants (60 participants in group A and 20 in group B). At day 28, three study participants had abandoned the study for personal reasons and therefore, both immunogenicity and safety data of these participants are missing after day 28 (one of low dose group and two of high dose group). This information was already described in the Methods section of the original manuscript and in the corresponding diagram (Figure 1).

In addition, one study participant was tested as SARS-CoV-2 positive in the fifth visit of the protocol (day 28 after first dose). The participant was completely asymptomatic and following protocol instructions, the application of the second dose was delayed. While the safety data of this volunteer at 28 days and after second booster are included, the immunogenicity data at day 28 and later time points were excluded

because the immunogenicity against the virus may shape the antibody response and lead to misinterpretation of the results. Therefore, Figures and tables with data of 28 days after first dose or longer had data of 58 and 18 volunteers. As the reviewer suggested we added the N of participants in each group in each Figure.

We added the following description in the study description. Page 16, lines 333-338:

“One study participant was tested as SARS-CoV-2 positive in the fifth visit of the protocol (day 28 after first dose). The participant was completely asymptomatic and following protocol instructions the application of the second dose was delayed. While the safety data of this volunteer at 28 days and after the second booster are included, the immunogenicity data at day 28 and at later time points were excluded because the immunogenicity against the virus may shape the antibody response and lead to misinterpretation of the results.”

7) Line 331 states "Nonparametric statistical tests were used to analyze titer data" but I did not seem the used for categorical variables. I think this sentence can be deleted since there is a better description later in the paragraph.

Replay: Following the reviewer’s suggestion we deleted the sentence.

8) Time since infection is quite different between two dose groups. This probably should be mentioned. Also, the percentage with previous covid seems to be quite a bit larger for the higher dose level. Similarly, the mRNA cohort has no participants with prior COVID-19. Please add these issues to the discussion.

Replay: We added in the discussion this limitation, Page 12, Lines 229-233:

“Although the time since prior COVID-19 declared by participants was quite different between the high and low dose groups and this could be a limitation, the analysis of anti-N in sera of all individuals indicated that both populations were similar in their inferred previous exposure to the virus and that the immune responses after vaccination are independent of previous infection status.”

Regarding the mRNA cohort, while all subjects of this cohort declared no previous infection, serological analysis of anti-N antibodies indicated that half of them have had previous contact with the virus. This proportion is similar to that observed in group A and group B. We added and discussed this limitation in the discussion of the manuscript, Page 14, lines 273-276:

“The confirmed previous SARS-CoV-2 infection history (0% in group BNT16b2 versus 8% and 20% in groups A and B, respectively) was quite different, nevertheless anti-N antibodies serology indicated that the three populations had similar proportions of seropositive individuals and might had similar previous contact with the virus.”

References

- 1 Liu, X. *et al.* Persistence of immunogenicity after seven COVID-19 vaccines given as third dose boosters following two doses of ChAdOx1 nCov-19 or BNT162b2 in the UK: Three month analyses of the COV-BOOST trial. *J Infect* **84**, 795-813, doi:10.1016/j.jinf.2022.04.018 (2022).
- 2 Zhang, Y. *et al.* Immunogenicity, durability, and safety of an mRNA and three platform-based COVID-19 vaccines as a third dose following two doses of CoronaVac in China: A randomised, double-blinded, placebo-controlled, phase 2 trial. *EClinicalMedicine* **54**, 101680, doi:10.1016/j.eclinm.2022.101680 (2022).
- 3 Canetti, M. *et al.* Immunogenicity and efficacy of fourth BNT162b2 and mRNA1273 COVID-19 vaccine doses; three months follow-up. *Nat Commun* **13**, 7711, doi:10.1038/s41467-022-35480-2 (2022).

Reviewers' Comments:

Reviewer #2:

Remarks to the Author:

The authors have responded well to all of my major concerns. In my opinion at least - they also appear to have responded reasonably to the concerns expressed by the other reviewers.

In their response to Reviewer #1, they also make the point that this work is highly unusual because it represents the effort of a scientist in a South American country: something to be quite proud of in the context of globalization of pandemic preparedness.

The additional data and analyses provided by the authors are very useful in interpreting their results. Although it is relatively late in the pandemic and the Gamma variant is 'long-gone' ... it is nonetheless quite informative to see the degree of cross-reactivity induced by a pure Gamma boost, including against some of the most recently-emerged variants.

Reviewer #3:

Remarks to the Author:

My only comment on this revision is that in the statistics section there is a sentence that states no assumptions on missing data were made. This sentence should be deleted. The authors should state that missing data are assumed to be missing at random. This is because the analysis only includes observed data and ignores missing data.

Point-by-point response to the reviewers' comments

We thank again the reviewers for the valuable comments, which in our opinion have further improved our manuscript.

Please find below the point-by-point response to the reviewers' comments. Reviewers' comments are written in black. Responses in blue.

REVIEWERS' COMMENTS

Reviewer #2 (Remarks to the Author):

The authors have responded well to all of my major concerns. In my opinion at least - they also appear to have responded reasonably to the concerns expressed by the other reviewers.

In their response to Reviewer #1, they also make the point that this work is highly unusual because it represents the effort of a scientist in a South American country: something to be quite proud of in the context of globalization of pandemic preparedness.

The additional data and analyses provided by the authors are very useful in interpreting their results. Although it is relatively late in the pandemic and the Gamma variant is 'long-gone' ... it is nonetheless quite informative to see the degree of cross-reactivity induced by a pure Gamma boost, including against some of the most recently-emerged variants.

Reply: We thank the reviewer for the appreciation of our work and for the critical review that has impacted in the improvement of the final version of the manuscript.

Reviewer #3 (Remarks to the Author):

My only comment on this revision is that in the statistics section there is a sentence that states no assumptions on missing data were made. This sentence should be deleted. The authors should state that missing data are assumed to be missing at random. This is because the analysis only includes observed data and ignores missing data.

Reply: We thank the reviewer for this correction in the statistical section. As suggested by the reviewer we deleted the sentence that stated "no assumptions on missing data were made" and stated instead that "missing data are assumed to be missing at random".